# Ocean-E2E: Hybrid Physics-Based and Data-Driven Global Forecasting of Marine Heatwaves with End-to-End Neural Assimilation

## Abstract

This work focuses on the end-to-end forecast of global extreme marine heatwaves (MHWs), which are unusually warm sea surface temperature events with profound impacts on marine ecosystems. Accurate prediction of extreme MHWs has significant scientific and financial worth. However, existing methods still have certain limitations in forecasting general patterns and extreme events. In this study, to address these issues, based on the physical nature of MHWs, we created a novel hybrid data-driven and numerical MHWs forecast framework Ocean-E2E, which is capable of 40-day accurate MHW forecasting with end-to-end data assimilation. Our framework significantly improves the forecast ability of MHWs by explicitly modeling the effect of oceanic mesoscale advection and air-sea interaction based on a dynamic kernel. Furthermore, Ocean-E2E is capable of end-to-end MHWs forecast and regional high-resolution prediction, allowing our framework to operate completely independently of numerical models while outperforming the current state-of-the-art ocean numerical/AI forecasting-assimilation models. Experimental results show that the proposed framework performs excellently on global-to-regional scales and short-to-long-term forecasts, especially in those most extreme MHWs. Overall, our model provides a framework for forecasting and understanding MHWs and other climate extremes.

## 1 Introduction

Marine heatwaves (MHWs) are abnormally warm seawater events that significantly damage marine ecosystems Oliver et al. (2021); Pearce et al. (2011). In other words, MHWs are extreme **Sea Surface Temperature Anomaly (SSTA)** events. MHWs, particularly extreme ones, can cause coral bleaching Hughes et al. (2017; 2018) and widespread mortality of marine organisms Garrabou et al. (2009); Thomson et al. (2015). As a result, precise forecasting of extreme MHWs has significant scientific and economic implications. For example, synoptic scale MHWs forecasting can help seafood production and management planning, such as feed cycles, at 1-7 day timescales, while subseasonal to seasonal forecasting can further support proactive decision-making for the blue economy Hobday et al. (2016); Malick et al. (2020); Mills et al. (2017); Payne et al. (2022). In this study, we will mainly focus on the subseasonal-to-seasonal forecast (i.e., 1-40 days).

Traditional MHW forecasting has primarily relied on two paradigms: physics-based numerical models and emerging data-driven approaches. Numerical models solve oceanic primitive equations to produce seasonal forecasts that capture large-scale patterns of MHW onset and intensification Jacox et al. (2022); Brodie et al. (2023), as well as sub-seasonal predictions that resolve finer-scale variability over shorter lead times Benthuysen et al. (2021); Yu et al. (2024). In parallel, recent advances in deep learning have shown promise for efficient global ocean forecasting Bi et al. (2022); Chen et al. (2023a); Kurth et al. (2023), with extensions to MHW prediction Lin et al. (2023); Sun et al. (2023). Despite their operational value, both paradigms suffer from inherent limitations:

1) **Numerical models**: These approaches demand substantial computational resources Xiong et al. (2023); Wang et al. (2024), especially in the data assimilation processes. Moreover, they often parameterize key physical processes—such as air-sea coupling and mesoscale eddy dynamics—through

empirical formulations, leading to systematic biases and constrained skill in predicting the general patterns of MHWs Jacox et al. (2022); Giamalaki et al. (2022).

2) **Data-driven models**: While these methods statistically approximate complex oceanic dynamics, they frequently generate non-physical fields, exhibiting limited accuracy in forecasting extreme events like MHWs due to issues such as error accumulation and smoothing of SST gradients during autoregressive rollouts Wang et al. (2024). Additionally, they typically depend on initialization from numerical models, precluding end-to-end operation from raw observational inputs Xiong et al. (2023).

Given these complementary strengths and weaknesses, a natural question arises: *why not integrate the advantages of both by combining a relatively simple physical model with AI components, where the physical model handles well-understood dynamics and AI simulates the more complex or uncertain aspects?* In this study, we pursue this hybrid strategy to effectively mitigate the aforementioned drawbacks of numerical and data-driven models, enabling our framework to harness the best of both worlds:

1) **Enhanced computational efficiency and end-to-end capability**: By substituting neural networks for the forecasting and data assimilation processes, we have substantially reduced the cost of predicting MHWs, enabling our framework to operate independently of numerical models.

2) **Improved physical realism and reduced biases**: The incorporation of physical constraints ensures generation of consistent fields, addressing the non-physical artifacts and extreme-event limitations in pure data-driven methods.

By addressing these integrated shortcomings, we introduce Ocean-E2E, a global hybrid physics-based data-driven framework that significantly enhances forecasts of extreme MHWs, drawing inspiration from numerical models. The contributions of this paper can be summarized as follows: (1) **Global/Regional MHWs Forecasting Framework**. We propose a hybrid physics-based data-driven method that supports both global scale and regional high-resolution forecasting, achieving high-accuracy results for extreme MHWs forecasts. (2) **Global MHWs Neural Assimilation Framework**. We designed an end-to-end MHWs neural system that, when combined with the forecasting model, can directly obtain the complete initial field and forecasting results of MHWs from observation data. (3) **Physics-consistent Forecast**. By combining physical laws into our framework, our model has achieved state-of-the-art results in long-term and regional high-resolution forecasting with higher physical consistency.

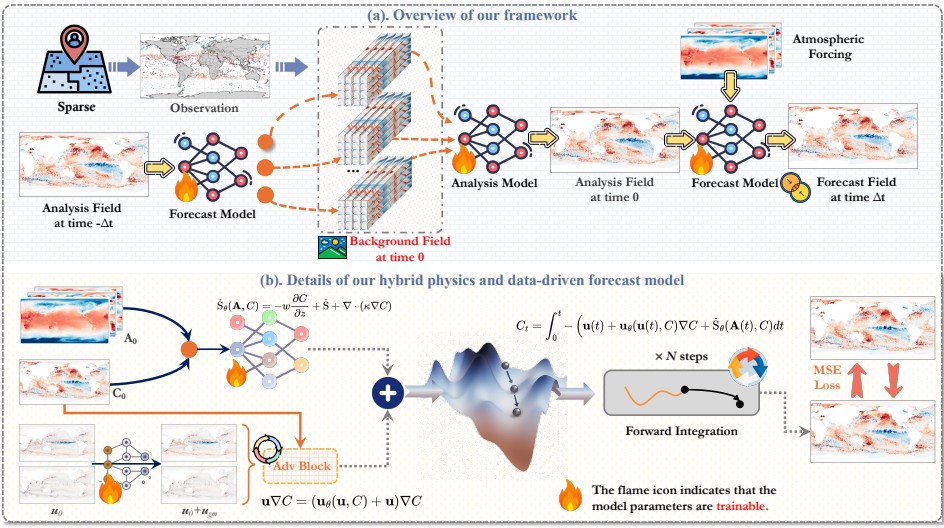

Figure 1: Our proposed Ocean-E2E framwork. a) Overview of our framework. b) Details of our hybrid physics and data-driven forecast model.

## 2 METHOD

### 2.1 FRAMEWORK OVERVIEW

In this study, the problem of the end-to-end MHWs forecast is **essentially forecasting SSTA**, which can be separated into two parts (shown in Figure 1). First, estimating the current state of SSTA $C_0 \in \mathbb{R}^{H \times W}$ (a global two-dimensional field, where $H$ and $W$ correspond to latitude and longitude) based on the observation $C_0^{obs}$ and the background field $C_0^{bg}$. Second, predicting the future state of mixed layer SSTA $C_t$ based on the current state $C_0$ under atmospheric forcing $A_{0:t} \in \mathbb{R}^{C \times H \times W}$. Here $A_{0:t}$ denotes the atmosphere state from time 0 to $t$. Formally speaking, we are going to model the conditional distribution $p(C_t | C_0^{obs}, C_0^{bg}, A_{0:t},)$ in two stages:

$$p(C_t | C_0^{obs}, C_0^{bg}, A_{0:t}) = \underbrace{p^f(C_t | C_0, A_{0:t})}_{\text{Forecast model}} \cdot \underbrace{p^a(C_0 | C_0^{obs}, C_0^{bg})}_{\text{Assimilation model}} \tag{1}$$

where $p^f$ and $p^a$ denote forecast and assimilation process, respectively. In our end-to-end framework, we model these two processes through two specifically designed neural networks $\phi_\theta^f$ and $\phi_\theta^a$, which we will discuss in detail in the following sections.

### 2.2 THE DESIGN OF $\phi_\theta^f$: HYBRID PHYSICS-AI MODEL

As established in the problem formulation, forecasting MHWs fundamentally reduces to predicting extreme sea surface temperature anomalies (SSTA). To effectively embed physical principles into our modeling framework, we must rigorously account for the governing laws of SSTA evolution. We approach this challenge through a generalized geophysical fluid dynamics perspective, wherein seawater temperature constitutes a *passive tracer* advected by oceanic flow fields. Formally speaking, the governing equation of SSTA $C$, together with other surface variables can be written as:

$$\frac{\partial C}{\partial t} + (\mathbf{u}_g + \mathbf{u}_{ag})\nabla C + w\frac{\partial C}{\partial z} = \dot{S} \tag{2}$$

$$f\mathbf{k} \times \mathbf{u}_g = -\nabla(g\eta) \tag{3}$$

$$\frac{\partial \eta}{\partial t} + \nabla \cdot (H\hat{\mathbf{u}}) = 0 \tag{4}$$

$$\frac{\partial \hat{\mathbf{u}}}{\partial t} + \widehat{\mathbf{G}_u} + \nabla(g\eta) = 0 \tag{5}$$

$$w|_{z=0} = \eta_t \tag{6}$$

Where $t$ is time, $(\mathbf{u}, w) = (\mathbf{u}_g + \mathbf{u}_{ag}, w)$ are horizontal and vertical flow velocity ($\mathbf{u}_g$ represents oceanic geostrophic velocity while $\mathbf{u}_{ag}$ is ageostrophic velocity), $\nabla$ is the horizontal derivation, $\dot{S}$ is the sink/source, $\mathbf{k}$ is a unit vector in the vertical direction, $f$ is the Coriolis parameter, $\eta$ is sea surface height, $H$ is ocean depth, $\hat{\cdot}$ is vertical average operator (i.e., $\hat{\mathbf{u}} = \int_{-H}^0 \mathbf{u}dz$), $g$ is the gravitional acceleration. By integrating equation (1), the state of $C$ at time $t$ can be expressed in the form of the initial condition $C_0$:

$$\begin{aligned} C_t &= \int_0^t -\mathbf{u}_g\nabla C + (-\mathbf{u}_{ag}\nabla C - w\frac{\partial C}{\partial z} + \dot{S})\, dt + C_0 \\ &\simeq \int_0^t \underbrace{-\mathbf{u}_g\nabla C}_{\text{advective transport}} + \underbrace{(-\mathbf{u}_{ag}\nabla C + \dot{S})}_{\text{mixing and external sink/source}} \, dt + C_0 \end{aligned} \tag{7}$$

The second equal sign is due to the fact that, according to equation (5), $w|_{z=0}$ is small ($10^{-6} \sim 10^{-5}$ m/s) compared to other terms. Equation (7) implies two key governing mechanisms of MHWs: 1) *advective transport* by geostrophic currents that redistribute thermal energy, 2) *mixing and external sink/source* encompassing convective mixing and external ageostrophic forcing through surface boundary interactions. In the following paragraphs, we will combine physical theory and deep learning methods to simulate these two mechanisms in four steps.

***Step 1. Modelling advective transport***. The key factor of advective transport is the ocean surface current $\mathbf{u}_g$, where the flow field at the smaller scale (mesoscale) plays an important role in the distribution of MHWs and other oceanic passive tracers. However, in most models and data, due to limited resolution, the model cannot distinguish these small-scale processes, so the advection processes caused by the flow field below the model's finest grid (i.e., subgrid processes):

$$\overline{\mathbf{u}_g \nabla C} = \overline{\mathbf{u}_g}\overline{\nabla C} + \overline{\mathbf{u}_g' \nabla C'} \tag{8}$$

are not fully simulated. Here, the overline $\overline{\cdot}$ represents the downsampling operator. In other words, the variables with an overline denote the low-resolution data. In the traditional numerical model, Gent and McWilliams Gent & Mcwilliams (1990) found that this subgrid process can be fitted by 'bolus velocity' (also called GM90 parameterization):

$$\overline{\mathbf{u}_g' \nabla C'} \sim \mathbf{u}_{gm}\overline{\nabla C} \tag{9}$$

According to the GM90 parameterization, these bolus velocity can be expressed as Gent & Mcwilliams (1990):

$$\mathbf{u}_{gm} = (\kappa \nabla \rho / \rho_z)_z \tag{10}$$

where $\rho = \rho(C)$ is the density. It should be noticed that this expression requires subsurface ocean data, which is often extremely difficult to obtain. What's more, the choice of coefficient $\kappa$ requires an empirical formula, which is always inaccurate. Alternatively, we seek a neural network $\mathbf{u}_\theta(\mathbf{u}_g, C) = \mathbf{u}_{gm}$ to approximate the bolus velocity and the subgrid advection effect. In summary, the advection term can be expressed as:

$$\mathbf{u}_g \nabla C = (\mathbf{u}_\theta(\mathbf{u}_g, C) + \mathbf{u}_g)\nabla C \tag{11}$$

***Step 2. Modelling mixing and external sink/source***. This term is strongly related to the air-sea interaction along the ocean's surface. To be specific, for SSTA, the source term consists of four parts:

$$\dot{S} \propto Q'_{\text{net}} = Q'_{\text{SW}} + Q'_{\text{LW}} + Q'_{\text{sh}} + Q'_{\text{lh}} \tag{12}$$

where $Q_{\text{SW}}, Q_{\text{LW}}, Q_{\text{sh}}, Q_{\text{lh}}$ are the shortwave radiation, longwave thermal radiation, sensible heat flux, and latent heat flux, respectively. Of the four air-sea heat fluxes, the sensible and latent flux $Q'_{\text{sh}} + Q'_{\text{lh}}$ determines the majority of the SSTA's variations Holbrook et al. (2019). According to the bulk formula Cui et al. (2025), these two terms are significantly related to surface wind speed $U_{10}, V_{10}$, near-surface temperature of the atmosphere $T_{2m}$, and surface specific humidity $q_a$. What's more, the ageostrophic velocity $\mathbf{u}_{ag}$ is also driven by surface wind forcing $U_{10}, V_{10}$. Based on the observation above, we assume that mixing and external sink/source can be well approximated by the surface variables of the atmosphere $\mathbf{A} = (U_{10}, V_{10}, T_{2m}, q_a)$ and $C$ itself. Specifically,

$$\dot{S}_\theta(\mathbf{A}, C) = -\mathbf{u}_{ag}\nabla C + \dot{S} \tag{13}$$

Since this study primarily focuses on proposing a physics-AI **framework**, the design and selection of the specific structures for the two neural networks $\mathbf{u}_\theta, \dot{S}_\theta$ are highly flexible. In this study, we selected a powerful and robust spatiotemporal prediction backbone Tan et al. (2022) to accomplish this task. We believe that more advanced backbones can further enhance the model's prediction performance.

***Step 3. Treating boundary conditions***. Above paragraphs point out that in order to effectively model the key physical mechanism of MHWs, we need the future state of $\mathbf{A}$ and $\mathbf{u}_g$ (i.e., boundary conditions). For oceanic geostrophic velocity $\mathbf{u}_g$, $\frac{\partial}{\partial t}(3) - \nabla \cdot (H(4))$ implies that

$$\frac{\partial^2 \eta}{\partial t^2} + \nabla \cdot (gH\nabla\eta) = \nabla \cdot \widehat{\mathbf{G}_u} \simeq \nabla \cdot \widehat{\mathbf{G}_{u_g}} \triangleq \mathcal{G}(\eta) \tag{14}$$

which is a self-consistent hyperbolic equation. This indicates that, at least at a relatively short time interval, the evolution of $\eta$, together with geostrophic velocity, can be modelled based on an auto-regressive manner:

$$\frac{\partial \mathbf{u_g}}{\partial t} = \mathcal{M}_\theta(\mathbf{u_g}) \tag{15}$$

Here $\mathcal{M}_\theta$ is a pretrained neural network. Inspired by the Wu et al. (2024b) architecture, the neural network module $\mathcal{M}_\theta$ used for modeling ocean surface geostrophic velocity employs a U-Net-like encoder-decoder structure. This model is primarily built upon the **Group Attention Block**

(**GABlock**), which serves as the core computational unit. The GABlock is designed to efficiently capture multi-scale spatio-temporal features, leveraging a **Spatial Attention (SA)** module that incorporates depth-wise and depth-wise dilated convolutions for spatial gating, followed by a **Multilayer Perceptron (Mlp)**. The overall architecture processes initial geostrophic current conditions, uses downsampling and stacked GABlocks in the encoder to extract features, and then symmetrically reconstructs the resolution in the decoder via upsampling and skip connections to predict the future geostrophic velocity field. What's more, the atmosphere variables between time interval $[0, t]$ is also acquired through a large AI-driven weather forecast model Gao et al. (2025):

$$\frac{\partial \mathbf{A}}{\partial t} = \mathcal{N}_\theta(\mathbf{A}) \tag{16}$$

***Step 4. Combining physics and AI together***. As shown in the section above, the furture state of $C$ at time $t$ can be modelled as:

$$C_t = \int_0^t - \left[ \int_0^s \mathcal{M}_\theta(\mathbf{u}_g)ds + \mathbf{u}_\theta\big(\int_0^s \mathcal{M}_\theta(\mathbf{u}_g)ds, C\big) \right] \nabla C \\ + \dot{\mathrm{S}}_\theta\big(\int_0^s \mathcal{N}_\theta(\mathbf{A})ds, C\big)ds \tag{17}$$

In the training stage, we first pretrain these two neural networks $\mathcal{M}_\theta, \mathcal{N}_\theta$. Then the parameters of these networks are frozen and we utilize the forecast model to optimize the parameters of $\mathbf{u}_\theta, \dot{\mathrm{S}}_\theta$ based on the MSE loss, which can be written as:

$$\mathcal{L} = ||\hat{C}_t - \int_0^t -(\mathbf{u}_g + \mathbf{u}_\theta(\mathbf{u}_g, C))\nabla C + \dot{\mathrm{S}}_\theta(\mathbf{A}, C)dt||^2 \tag{18}$$

Where $\hat{C}_t$ is the groundtruth data of $C_t$. In practice, by separating the time interval $[0, t]$ into $N$ subintervals $[i\Delta t, (i + 1)\Delta t], 1 \leq i \leq N - 1$, equation (18) is approximated via forward Euler method:

$$C_{(i+1)\Delta t} = C_{i\Delta t} + \Big[ - \big(\mathbf{u}_{i\Delta t} + \mathbf{u}_\theta(\mathbf{u}_{i\Delta t}, C_{i\Delta t})\big)\nabla C_{i\Delta t} \\ + \dot{\mathrm{S}}_\theta(\mathbf{A}_{i\Delta t}, \mathbf{A}_{(i+1)\Delta t}, C_{i\Delta t}) \Big]\Delta t, \tag{19}$$

$$\mathbf{u}_{(i+1)\Delta t} = \mathbf{u}_{i\Delta t} + \mathcal{M}_\theta(\mathbf{u}_{i\Delta t})\Delta t, \tag{20}$$

$$\mathbf{A}_{(i+1)\Delta t} = \mathbf{A}_{i\Delta t} + \mathcal{N}_\theta(\mathbf{A}_{i\Delta t})\Delta t. \tag{21}$$

Then the parameters of $\mathbf{u}_\theta, \dot{\mathrm{S}}_\theta$ are optimized through $\mathcal{L} = ||\hat{C}_t - C_{N\Delta t}(\theta)||^2$. It is worth noting that the above numerical kernel is highly **lightweight**, imposing no significant computational burden on our framework. Details of the numerical implementation can be found in Appendix C.

## 2.3 THE DESIGN OF $\phi_\theta^a$: NEURAL DATA ASSIMILATION

As shown in the above section, our hybrid AI-physics model requires an initial condition $C_0$ to initialize our framework. However, under realistic operational forecasting scenarios, the full $C_0$ (analysis field) cannot be directly observed. This necessitates a data assimilation (DA) process that optimally combines sparse observational data $C_0^{obs}$ with prior background estimates $C_0^{bg}$ from numerical models, which requires large computational resources Boudier et al. (2023).

Our framework addresses these limitations by establishing a **direct mapping** from observational inputs $C_0^{obs}$ to assimilated states $C_0$ through deep learning. We utilized neural networks to optimally combine sparse observations with background fields in one step. Specifically, we begin with the SSTA state at time $-\Delta t$ ($C_{-\Delta t}$), defining $[-\Delta t, 0]$ as the assimilation window. We generate background fields ($C_0^{bg}$) by adding structured noise to $C_{-\Delta t}$:

$$C_0^{bg} = \phi_\theta^f(C_{-\Delta t} + \varepsilon, \mathbf{A}_{0:\Delta t}) \tag{22}$$

where $\varepsilon$ denotes Perlin noise. The final analysis field is then produced by merging observations with background data:

$$C_0 = \phi_\theta^a(C_0^{obs}, C_0^{bg}) \tag{23}$$

where $\phi_\theta^a$ is an edge-reparameterization and attention-guided network with Kirsch-guided reparameterization, which is capable of reconstructing SSTA fields from sparse observations Liu et al. (2024). The neural assimilation network $\phi_\theta^a$, inspired by Liu et al. (2024), is an edge-reparameterization and attention-guided architecture meticulously designed for the efficient, real-time projection of observation data to analysis data. The network's core innovation lies in its Kirsch-guided Reparameterization Module (KRM), which efficiently consolidates standard, expanded, and eight directional Kirsch edge-detection convolutions into a single unit during inference, enabling robust capture of sharp feature boundaries. Complementing this, the network employs a joint Channel and Spatial Attention mechanism (CAM and SAM) to dynamically focus computational resources on crucial data regions. The final architecture is a cascaded, iterative structure built upon a Basic Block that includes the attention modules, where the observation data is passed through five consecutive stages, each integrating the KRM output with a residual connection to progressively refine the analysis and produce the highly accurate, edge-aware output. The details of $\phi_\theta^a$ can be found in Appendix D.

Once the initial analysis field $C_0$ is obtained, the model proceeds with forward assimilation over subsequent time steps to maintain accuracy in evolving forecasts. Starting from $C_0$, the framework iteratively advances the state by generating a background estimate for the next time step $C_{\Delta t}^{bg}$ using the forward operator and incorporating dynamical updates at a time interval $\Delta t$(i.e., assimilation window). For instance, at $i$-th time step, the background is computed as:

$$C_{(i+1)\Delta t}^{bg} = \phi_\theta^f(C_{i\Delta t} + \varepsilon_i, A_{i\Delta t:(i+1)\Delta t}) \tag{24}$$

followed by the assimilation step:

$$C_{(i+1)\Delta t} = \phi_\theta^a(C_{(i+1)\Delta t}^{obs}, C_{(i+1)\Delta t}^{bg}) \tag{25}$$

As we will demonstrate in the following paragraph, this autoregressive process can generate robust assimilation fields over hundreds of days.

## 3 EXPERIMENTS

We designed comprehensive experiments to evaluate our model. Our evaluation is guided by the following key questions: **RQ1: Overall Performance:** Can Ocean-E2E outperform state-of-the-art models with higher consistency and better performance in normal and extreme MHW events? **RQ2: End-to-end Forecasting:** Can Ocean-E2E run independently of numerical model while preserving a high accuracy in real-world MHWs forecast? **RQ3: High-resolution Multiscale Prediction:** Can our framework achieve a satisfying performance in regional high-resolution simulations, where complicated oceanic multi-scale dynamics pose more challenges to the model's forecast skill and physical consistency?

### 3.1 BENCHMARKS AND BASELINES

We conduct the experiments on GLORYS12V1 reanalysis data Lellouche et al. (2018), which provides daily mean data of sea surface temperature (SST) covering latitudes between -80° and 90° spanning between 1993 and 2021. The seasonal cycle has been removed from the original data in order to get the MHWs (SSTA) field Shu et al. (2025). The subset we use includes years from 1993 to 2021, which is 1993-2018 for training, 2019 for validating, and 2020-2021 for testing. The surface velocity of the ocean is obtained from satellite observation Pujol et al. (2016). The atmospheric variables are obtained from ECWMF Reanalysis v5 (ERA5) Hersbach et al. (2020). More details, including observational data and atmospheric forcing data, can be found in Appendix B.

Before introducing our experiments, there are two key concepts that should be clarified: **ocean simulation** and **end-to-end ocean forecast**. As pointed out in previous studies Shu et al. (2025);

Table 1: In the global ocean simulation task, we compare the performance of our Ocean-E2E with various baselines. The average results for global SSTA (Sea Surface Temperature Anomaly) are measured using RMSE and CSI. Lower RMSE (↓) and higher CSI (↑) indicate better performance. The best results are in **bold**, and the second-best are underlined. Note that some baseline models are unstable and produce unrealistic forecasts, with RMSE exceeding 10 K replaced by —.

| Model Category | Metric | | | | | | | |
|---|---|---|---|---|---|---|---|---|
| | 20-day | | 40-day | | 50-day | | 60-day | |
| | RMSE | CSI | RMSE | CSI | RMSE | CSI | RMSE | CSI |
| **Ocean Forecasting Models** | | | | | | | | |
| ☁ FourCastNet Pathak et al. (2022) | 0.6836 | 0.2709 | — | 0.1755 | — | 0.1537 | — | 0.1423 |
| ☁ CirT Liu et al. (2025) | 1.3496 | 0.0905 | 1.9249 | 0.0810 | 1.9932 | 0.0770 | 2.0321 | 0.0746 |
| ☁ WenHai Cui et al. (2025) | **0.5435** | 0.4202 | 0.7006 | 0.2805 | 0.7633 | 0.2447 | 0.8139 | 0.2209 |
| ☁ ClimODE Verma et al. (2024) | 0.7221 | 0.3263 | 0.8555 | 0.2091 | 0.8997 | 0.1896 | — | 0.1754 |
| **Operator Learning Models** | | | | | | | | |
| ☘ CNO Raonic et al. (2023) | 0.7062 | 0.3669 | 0.9025 | 0.2367 | 0.9821 | 0.2011 | 1.0556 | 0.1771 |
| ☘ LSM Wu et al. (2023) | 1.1346 | 0.3025 | — | 0.1760 | — | 0.1436 | — | 0.1312 |
| **Computer Vision Backbones** | | | | | | | | |
| ▣ U-Net Ronneberger et al. (2015) | 0.6923 | 0.3909 | 0.8893 | 0.2629 | 0.9706 | 0.2292 | 1.0463 | 0.2054 |
| ▣ ResNet He et al. (2016) | — | 0.2537 | — | 0.1538 | — | 0.1248 | — | 0.1029 |
| ▣ DiT Peebles & Xie (2023) | 0.9390 | 0.3411 | 1.7051 | 0.2528 | 2.2474 | 0.2311 | 2.9139 | 0.2137 |
| **Spatiotemporal Models** | | | | | | | | |
| ▦ ConvLSTM Shi et al. (2015) | 0.7135 | 0.3585 | 0.8920 | 0.2292 | 0.9726 | 0.1957 | 1.0545 | 0.1735 |
| ▦ SimVP Tan et al. (2022) | 0.6729 | 0.3749 | 0.8345 | 0.2736 | 0.8864 | 0.2413 | 0.9296 | 0.2162 |
| ▦ PastNet Wu et al. (2024b) | 1.3876 | 0.1867 | 1.4230 | 0.1760 | 1.4287 | 0.1733 | 1.4353 | 0.1705 |
| 🏆 **Ocean-E2E** | 0.5659 | **0.4285** | **0.6596** | **0.3230** | **0.6911** | **0.2874** | **0.7194** | **0.2580** |
| Promotion | — | 2.0% | 5.8% | 15.1% | 9.5% | 17.4% | 11.6% | 16.8% |

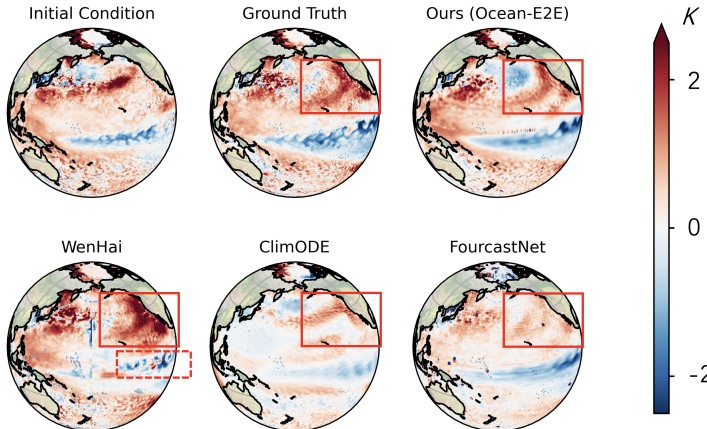

Figure 2: Snapshots of our framework and other baseline simulations at 60-day lead time. The simulations are initialized on August 7th, 2020. Red solid boxes represent MHW events while red dashed box indicates physical inconsistency in forecasting results.

Cui et al. (2025), the ocean simulation utilizes the current state of SSTA $C_0$ to forecast future state $C_t$ under the realistic atmosphere forcing $\mathbf{A}_{0:t}$. In contrast, the end-to-end forecast task includes assimilating the initial analysis field of SSTA $C_0$ and forecasting only based on the current atmosphere state $\mathbf{A}_0$.

We compare the simulation performance of Ocean-E2E with 4 main catergories of data-driven models: **ocean forecasting models** (Fourcastnet Kurth et al. (2023), CirT Liu et al. (2025), Wenhai Cui et al. (2025) and ClimODE Verma et al. (2024)), **operator learning models** (FNO Li et al. (2021), CNO

Table 2: In operational global ocean forecast task, we compare the performance of our Ocean-E2E with S2S. The average results for global SSTA of RMSE are recorded for 2020 and 2021. A small RMSE (↓) indicates better performance. The best results are in **bold**.

| MODEL | METRIC (RMSE) | | | | | | | |
|---|---|---|---|---|---|---|---|---|
| | 2020 | | | | 2021 | | | |
| | 10-DAY | 20-DAY | 30-DAY | 40-DAY | 10-DAY | 20-DAY | 30-DAY | 40-DAY |
| S2S | 0.8140 | 0.8870 | 0.9514 | 0.9965 | 0.8261 | 0.8897 | 0.9450 | 0.9895 |
| OCEAN-E2E (OURS) | **0.5750** | **0.6047** | **0.7514** | **0.8747** | **0.6414** | **0.7447** | **0.8147** | **0.8729** |
| OCEAN-E2E (PROMOTION) | 29.4% | 31.8% | 21.02% | 12.2% | 22.4% | 16.3% | 13.7% | 11.7% |

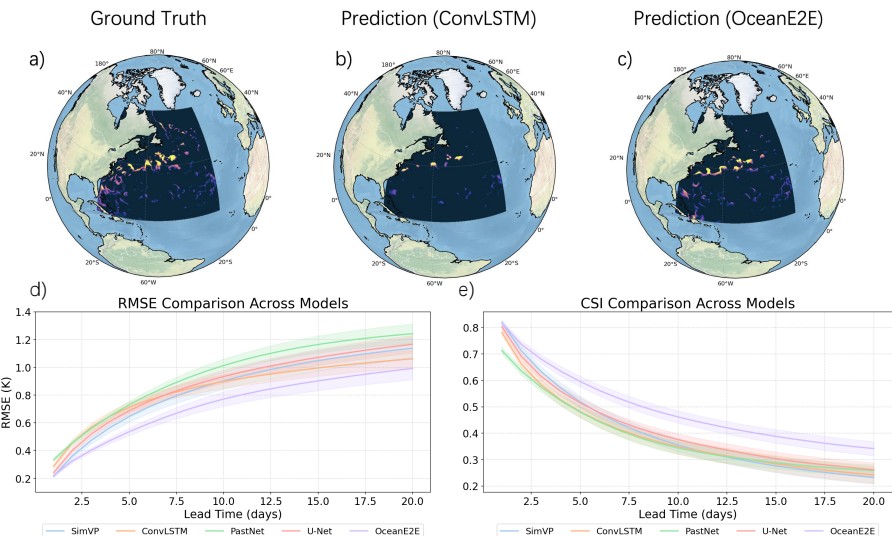

Figure 3: Results of our regional high resolution MHWs simulation in western Atlantic Ocean. a)-c) Snapshots of our simulation, where **yellow** parts indicate MHWs. d)-e) RMSE and CSI of our Ocean-E2E and other baselines.

Raonic et al. (2023) and LSM Wu et al. (2023)), **computer vision backbones** (U-Net Ronneberger et al. (2015), ResNet He et al. (2016) and DiT Peebles & Xie (2023)) and **spatiotemporal models** (ConvLSTM Shi et al. (2015), SimVP Tan et al. (2022) and PastNet Wu et al. (2024b)).

We also compare the end-to-end operational forecast ability against **traditional numerical operational forecast system**: ECMWF sub-seasonal to seasonal prediction (S2S). S2S is currently the superior subseasonal-to-seasonal weather forecast model that provides the forecast of atmosphere and ocean surface physics at a lead time of 40 days. The original resolution is 1.5°, and we upsample the data to 1/2° for comparison.

### 3.2 EVALUTION METRIC

We utilized two metrics, RMSE (Root Mean Square Error) and CSI (Critical Success Index), to evaluate the forecast performance. The RMSE represents the **overall performance** of our framework's forecast, while CSI mainly focuses on the most **extreme MHW events**. What's more, we utilize assimilation bias (BIAS) to assess the systematic bias induced by our neural assimilation system. More details can be found in Appendix E.

### 3.3 EVALUATION OF OCEAN SIMULATION PERFORMANCE (RQ1)

As shown in Table 1, it can be found that in long-term ocean simulations (40-60 days), our framework performs better compared to the baseline. On day 60, the RMSE decreased by an average of 11.6%

and the CSI increased by an average of 9.2%. In shorter simulations (20 days), our model was slightly lower than WenHai but significantly higher than other baselines. This indicates that our Ocean-E2E can capture both general patterns and extreme events of MHWs. To further illustrate our simulation ability, we select a snapshot of our 60 days simulation, as shown in Figure 2. It can be found that a strong MHW is growing in the West Pacific Ocean (red solid box in Figure 2). Compared to other baselines, our Ocean-E2E shows a robust forecast result and better consistency with ground truth.

### 3.4 EVALUATION OF THE END-TO-END NEURAL ASSIMILATION AND FORECAST PERFORMANCE (RQ2)

We conducted two sets of assimilation experiments using analysis fields from January 1, 2020 and January 1, 2021 as initial conditions, respectively. Both experiments employed a 6-day assimilation window to assimilate SSTA fields over a full year. As shown in Appendix G.1 (Figure 5), the RMSE and BIAS evolution reveal that assimilation errors rapidly increase and stabilize during the assimilation process, demonstrating the stability of our assimilation framework.

Using these assimilated fields as initial conditions and GLORYS12V1 reanalysis as ground truth, we evaluate the forecasting performance of Ocean-E2E against the S2S system. **Since S2S provides 46 days of forecast data, in this study, we only compare the 10-40 days' forecast results of Ocean-E2E with it**. To ensure experimental fairness, results in Table 2 were calculated after the assimilation outputs stabilized (12 days post-initialization). It can be found that Ocean-E2E achieves an average 10% reduction in RMSE compared to superior end-to-end numerical prediction systems across 40-day subseasonal-to-seasonal forecasts. Additional comparative experiments have been conducted using S2S assimilation fields as both reference truth and initial conditions (see Appendix G). In all experimental configurations, Ocean-E2E demonstrates superior performance relative to S2S, confirming its robust competitiveness.

### 3.5 REGIONAL HIGH-RESOLUTION SIMULATION (RQ3)

As discussed in Section 2.1, our framework builds upon the GM90 theoretical framework, which is also applicable to high-resolution data. To validate its performance, we implement a regional high-resolution simulation ($1/12°$) of the western Atlantic Ocean. Figure 3 a)-c) illustrate a comparison between Ocean-E2E and baselines. Compared to the ground truth, our model has more physical details with more extreme MHW events (yellow parts). Evaluation based on RMSE and CSI (Figure 3 d)-e)) also confirmed that Ocean-E2E achieves a state-of-the-art performance in regional simulations. Furthermore, to verify physical consistency, we calculated the power spectrum of the Ocean-E2E simulation results and compared it with the baselines (see Appendix G.2). The results demonstrate that our model exhibits higher consistency with the real physical fields at small scales (high frequencies).

Table 3: CSI of our ablation studies on model design, forcings, and neural architecture. The best results are in **bold**.

| Variants | Global Simulation | | Regional Simulation | |
|---|---|---|---|---|
| | 20 day | 60 day | 10 day | 30 day |
| *Physical Model Design* | | | | |
| $\dot{S}_\theta$ (S only) | 0.3884 | 0.1658 | 0.3913 | 0.1877 |
| $(\mathbf{u}_\theta + \mathbf{u}_g)\nabla C$ (ADV only) | 0.3207 | 0.1504 | 0.3207 | 0.0963 |
| $\mathbf{u}_g \nabla C$ (pure numerical) | — | — | — | — |
| *Time-Varying Forcings* | | | | |
| Static Atmosphere ($\mathbf{A}(t) \leftarrow \mathbf{A}(0)$) | 0.2652 | 0.1223 | 0.3021 | 0.1470 |
| Static Background Flow ($\mathbf{u}_g(t) \leftarrow \mathbf{u}_g(0)$) | 0.3988 | 0.2115 | 0.4080 | 0.2245 |
| *Neural Architecture Analysis* | | | | |
| w/o GABlock (in $\mathcal{M}_\theta$) | 0.4015 | 0.2245 | 0.4322 | 0.2588 |
| w/o MidXnet (in $\dot{S}_\theta, \mathbf{u}_\theta$) | 0.3540 | 0.1810 | 0.3890 | 0.2015 |
| $(\mathbf{u}_\theta + \mathbf{u}_g)\nabla C + \dot{S}_\theta$ (Ours) | **0.4285** | **0.2580** | **0.4615** | **0.2926** |

## 3.6 ABLATION STUDIES

To validate the effectiveness of the proposed framework, we perform an ablation analysis by se-quentially removing core components in our global and regional simulation task. Since our main contribution is the framework rather than neural network design, we choose the two key components of our framework: the advection term $(\mathbf{u}_\theta(\mathbf{u}_g, C) + \mathbf{u}_g)\nabla C$ (denoted as ADV) and the source term $\dot{S}_\theta(\mathbf{A}, C)$ (denoted as S). As demonstrated in Table 3, the Critical Success Index (CSI) exhibits a marked reduction when either component is omitted, confirming their essential roles in enhancing simulation accuracy, particularly for extreme marine heatwaves (MHWs). Notably, the source term contributes most significantly to forecasting performance, underscoring the critical influence of air-sea interaction dynamics on MHW evolution. Moreover, if the neural networks are removed from the model (i.e., a pure numerical model), the model collapses quickly due to the numerical instability caused by large horizontal gradients in the advection terms. This stresses the importance of the GM 'bolus' velocity, which acts as a subgrid 'damping' term.

Furthermore, we investigated the impact of time-varying forcings by fixing the atmospheric forcing ($\mathbf{A}$) and background flow ($\mathbf{u}_g$) to their initial states. As shown in the second section of Table 3, using static atmospheric forcing results in a significant performance drop (e.g., Global 60-day CSI decreases to 0.1223), confirming that dynamic heat flux is critical for SSTA prediction. Similarly, fixing the background flow degrades performance, indicating the necessity of capturing the time-varying nature of large-scale ocean circulation.

Finally, we analyzed the contribution of specific neural network components. We examined the Group Attention Block (`GABlock`) in the background modeling branch and the `MidXnet` in the dynamical evolution branch. The results in the third section of Table 3 indicate that removing the `GABlock` leads to a noticeable decline in accuracy, verifying its importance in encoding background physics. Moreover, removing the `MidXnet` causes a substantial drop in performance, validating its critical role in capturing complex spatiotemporal dynamics in the latent space.

## 4 CONCLUSIONS

In this study, we introduce Ocean-E2E, a hybrid framework merging data-driven and numerical methods for end-to-end marine heatwave (MHW) forecasting. By resolving mesoscale advection and air-sea interactions with a dynamic kernel, it delivers stable 40-day global-to-regional MHW predictions in both simulation and forecast tasks. Neural assimilation enables independent high-resolution forecasting with enhanced accuracy and stability, via rapid error stabilization. Results confirm its robustness across scales, advancing MHW prediction and climate insights.

## ETHICS STATEMENT

This work presents a framework for simulating ocean dynamics and does not raise any ethics concerns.

## REPRODUCIBILITY STATEMENT

To guarantee the reproducibility of our results, the complete source code and data associated with this study have been made publicly available. The implementation, which comprises the simulation framework, training procedures, and evaluation scripts, can be accessed via the following anonymous GitHub link: `https://anonymous.4open.science/r/Ocean-E2E-anonymous-5994`.

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

# A ALGORITHM

We summarize the overall framework of Ocean-E2E in Algorithm 1.

---

**Algorithm 1** Ocean-E2E Framework for Global MHWs Forecast

---

**Require:** Initial SSTA observation $C_t^{obs}$, analysis field $C_{t-1}^a$ (at time $t-1$), current state of atmospheric forcing $\mathbf{A}_t$ and ocean current velocity $\mathbf{O}_t$ .
**Ensure:** Current step SSTA analysis field $C_t^a$, Next step SSTA prediction $C_{t+1}$.
 1: Initialize Ocean-E2E
 2: **repeat**
 3:     **Data Assimilation**
 4:     Perturb the initial SSTA field $C_{t-1}^a$ with $N = 10$ Perlin noise $C_{t-1}^{a(i)} = C_{t-1}^a + \varepsilon^{(i)}, i = 1, 2, ..., N$.
 5:     Feeding the perturbed ensembles into our trained forecast model to generate forecast ensemble: $C_t^{bg(i)} = \phi_\theta^f(C_{t-1}^{a(i)}, \mathbf{A}_{t-1:t})$, which are also called background fields at time $t$.
 6:     Using the neural assimilation network $\phi_\theta^a$ to produce the analysis field at time $t$: $C_t^a = \phi_\theta^a(C_t^{bg(i)}, C_t^{obs})$
 7:     **Hybrid physics data-driven forecast**
 8:     Calculate the future atmospheric forcing and ocean current velocity: $\mathbf{A}_{t+1} = \mathcal{N}_\theta(\mathbf{A}_t), \mathbf{O}_{t+1} = \mathcal{M}_\theta(\mathbf{O}_t)$.
 9:     Taking the current analysis field $C_t^a$, and integrating the equation (17) with $\mathbf{A}_{t:t+1}$ and $\mathbf{O}_{t:t+1}$ to get the forecast field $C_{t+1}$
10: **until** converged
11: **return** $Ocean - E2E$

---

# B DATA DETAILS

## B.1 DATASET

In this section, we will introduce the dataset used in this study in detail.

**Data used to train $\mathcal{N}_\theta$.** As mentioned in Section 3.2, we need to pretrain an neural network $\mathcal{N}_\theta$ to model the evolution of the atmosphere forcing $\mathbf{A}$. In this study, we utilize one of the state-of-the-art weather forecast model: OneForecast Gao et al. (2025). The data used to train OneForecast is obtained from weatherbench2 Rasp et al. (2024) benchmark, which is a subset of ERA5 reanalysis data Hersbach et al. (2020). Details can be found in Table 4.

Table 4: The data used to train $\mathcal{N}_\theta$.

| VARIABLE NAME | LAYERS | SPATIAL RESOLUTION | DT | LAT-LON RANGE | TIME |
|---|---|---|---|---|---|
| GEOPOTENTIAL (Z) | 13 | 1.5° | 6H | GLOBAL | 1959~2021 |
| SPECIFIC HUMIDITY (Q) | 13 | 1.5° | 6H | GLOBAL | 1959~2021 |
| TEMPERATURE (T) | 13 | 1.5° | 6H | GLOBAL | 1959~2020 |
| U COMPONENT OF WIND (U) | 13 | 1.5° | 6H | GLOBAL | 1959~2021 |
| V COMPONENT OF WIND (V) | 13 | 1.5° | 6H | GLOBAL | 1959~2021 |
| 10 METRE U WIND COMPONENT (U10M) | 1 | 1.5° | 6H | GLOBAL | 1959~2021 |
| 10 METRE V WIND COMPONENT (V10M) | 1 | 1.5° | 6H | GLOBAL | 1959~2021 |
| 2 METRE TEMPERATURE (T2M) | 1 | 1.5° | 6H | GLOBAL | 1959~2021 |
| MEAN SEA LEVEL PRESSURE (MSLP) | 1 | 1.5° | 6H | GLOBAL | 1959~2021 |

**Data used to train $\mathcal{M}_\theta$.** As shown in Section 3.1, like the atmosphere forcing $\mathbf{A}$, we need another neural network $\mathcal{M}_\theta$ to model the ocean surface current dynamics. The data of surface current is obtained from satellie observation, which can be downloaded from `https://data.marine.copernicus.eu/product/SEALEVEL_GLO_PHY_L4_MY_008_047`. Details of this dataset can be found in Table 5.

Table 5: The data used to train $\mathcal{M}_\theta$.

| Variable Name | Spatial Resolution | DT | Lat-Lon Range | Time |
|---|---|---|---|---|
| Eastward Surface Geostrophic Current Velocity ($U_g$) | 0.25° | 24H | Global | 1993∼2021 |
| Northward Surface Geostrophic Current Velocity ($V_g$) | 0.25° | 24H | Global | 1993∼2021 |

**Data used to train the hybrid data-driven network**. The training of $\dot{S}_\theta$ and $\mathbf{u}_\theta$ involves three atmosphere variables and three ocean varibales, which can be found in Table 6. The SST dataset is selected from GLORYS12V1, which can be downloaded from `https://data.marine.copernicus.eu/product/GLOBAL_MULTIYEAR_PHY_001_030`. For regional high-resolution simulation, we choose a regional subset of GLORYS12V1 (Table 7).

Table 6: The data used to train $\dot{S}_\theta$ and $\mathbf{u}_\theta$ (global simulation).

| Variable Name | Spatial Resolution | DT | Lat-Lon Range | Time |
|---|---|---|---|---|
| 10 Metre U wind Component (U10M) | 0.25° | 24H | Global | 1993∼2021 |
| 10 Metre V wind Component (V10M) | 0.25° | 24H | Global | 1993∼2021 |
| 2 Metre Temperature (T2M) | 0.25° | 24H | Global | 1993∼2021 |
| Eastward Surface Geostrophic Current Velocity ($U_g$) | 0.25° | 24H | Global | 1993∼2021 |
| Northward Surface Geostrophic Current Velocity ($V_g$) | 0.25° | 24H | Global | 1993∼2021 |
| Sea Surface Temperature (SST) | 1/12° | 24H | Global | 1993∼2021 |

Table 7: The data used to train $\dot{S}_\theta$ and $\mathbf{u}_\theta$ (regional simulation).

| Variable Name | Spatial Resolution | DT | Lat-Lon Range | Time |
|---|---|---|---|---|
| U10M | 0.25° | 24H | $20 \sim 60°N, 30 \sim 80°W$ | 1993∼2021 |
| V10M | 0.25° | 24H | $20 \sim 60°N, 30 \sim 80°W$ | 1993∼2021 |
| T2M | 0.25° | 24H | $20 \sim 60°N, 30 \sim 80°W$ | 1993∼2021 |
| $U_g$ | 1/12° | 24H | $20 \sim 60°N, 30 \sim 80°W$ | 1993∼2021 |
| $V_g$ | 1/12° | 24H | $20 \sim 60°N, 30 \sim 80°W$ | 1993∼2021 |
| SST | 1/12° | 24H | $20 \sim 60°N, 30 \sim 80°W$ | 1993∼2021 |

**Observational data**. For observational data, we utlilize the Hadley Centre Integrated Ocean Database (sparse observations) and Global Ocean ODYSSEA L4 Sea Surface Temperature (satellite observations), which can be downloaded from `https://climatedataguide.ucar.edu/climate-data/hadiod-met-office-hadley-centre-integrated-ocean-database` and `https://data.marine.copernicus.eu/product/SST_GLO_PHY_L4_MY_010_044`, respectively.

### B.2 Data Preprocessing: Data Normalization

Given that this study takes into account the primitive equations of the ocean, to avoid disrupting physical laws through normalization, we do not apply normalization to any ocean variables (including SSTA, $U_g$, $V_g$) in this research. Instead, normalization is only applied to atmospheric variables (including U10M, V10M, T2M).

Significant magnitude disparities exist among different atmospheric variables. To prioritize predictive capability over inter-variable magnitude differences, we implemented standardized preprocessing through normalization. Statistical parameters (mean value $\mu$ and standard deviation $\sigma$) were derived from the 1991-2018 training period, with each climate variable being assigned independent scaling

coefficients. To be specific, the normalization procedure applied the transformation: $\mathbf{X}' = (\mathbf{X} - \mu)/\sigma$, where $\mathbf{X}$ denotes the raw input data.

### B.3 DATA PREPROCESSING: RESOLUTION ALIGNMENT

Since this study utilizes data from multiple sources with varying resolutions, alignment is required in both temporal and spatial dimensions. For temporal resolution, we directly combine daily averages and daily snapshots at the daily scale without special processing. For spatial resolution, we use bilinear interpolation (specifically, the `torch.nn.functional.interpolate` function) to interpolate data with different resolutions onto a unified grid of 1/2-degree (for global simulation) or 1/12-degree (for regional simulation). Additionally, for the sparse global in-situ observation data, we first filter out all sea surface observations and construct a 1/2-degree global grid. For each observation point, all corresponding grid cells within a 1.5°×1.5° area centered on the observation point are assigned the respective observation data.

## C DETAILS OF DYNAMICAL CORE OF OCEAN-E2E

In this section, we are going to introduce the design of our hybrid physics data-driven model. As shown in Section 3.2, the evolution equation of our hybrid physics data-driven framework can be written as:

$$C_t = \int_0^t -(\mathbf{u}_g(t) + \mathbf{u}_\theta(\mathbf{u}_g(t), C))\nabla C + \dot{S}_\theta(\mathbf{A}(t), C)dt + C_0 \tag{26}$$

**Spatial Discretization**. In this study, we utilized two-order center discretization to approximate $\nabla C$. To be sepcific,

$$\nabla C(i,j) = (\mathbf{D_x} * C(i,j), \mathbf{D_y} * C(i,j)) \tag{27}$$

Where and $*$ is convolution operator, $\mathbf{D_x}$ and $\mathbf{D_y}$ are horizontal gradients, which has the form

$$\mathbf{D_x} = \begin{pmatrix} 0 & 0 & 0 \\ -1/d_{i,j} & 0 & 1/d_{i,j} \\ 0 & 0 & 0 \end{pmatrix}, \mathbf{D_y} = \begin{pmatrix} 0 & 1/s_{i,j} & 0 \\ 0 & 0 & 0 \\ 0 & -1/s_{i,j} & 0 \end{pmatrix}, \tag{28}$$

$d_{i,j}$ and $s_{i,j}$ are longtitudal and latitdual grid scale at $(i, j)$.

What's more, unlike atmosphere variables, the ocean MHWs have coastline boundary, which should be treated properly to avoid any numerical instability. In our study, we mitigate this boundary effect through a boundary mask $\mathbf{M}$, where

$$\mathbf{M}(i,j) = \begin{cases} 0, & \text{if the grid cell is situated on land,} \\ & \text{or if any of its neighboring cells are on land} \\ 1, & \text{else} \end{cases} \tag{29}$$

Furthermore, a scaling factor $\varepsilon_{gm} = 0.1$ is employed on $\mathbf{u}_{gm}$ to avoid the instability of training. The equation 26 then becomes

$$C_t = \int_0^t \left[ -(\mathbf{u}_g(t) + \varepsilon_{gm}\mathbf{u}_\theta(\mathbf{u}_g(t), C)) \nabla C \right] \mathbf{M}$$
$$+ \dot{S}_\theta(\mathbf{A}(t), C) \, dt + C_0 \tag{30}$$

**Time Integration**. In this study, we employ the forward Euler method to approximate the time integration of the system, as described by the following discrete updates:

$$C_{(i+1)\Delta t} = C_{i\Delta t} + \left[ -(\mathbf{u}_{i\Delta t} + \mathbf{u}_\theta(\mathbf{u}_{i\Delta t}, C_{i\Delta t}))\nabla C_{i\Delta t} \right.$$
$$\left. + \dot{S}_\theta(\mathbf{A}_{i\Delta t}, \mathbf{A}_{(i+1)\Delta t}, C_{i\Delta t}) \right] \Delta t, \tag{31}$$

$$\mathbf{u}_{(i+1)\Delta t} = \mathbf{u}_{i\Delta t} + \mathcal{M}_\theta(\mathbf{u}_{i\Delta t})\Delta t, \tag{32}$$

$$\mathbf{A}_{(i+1)\Delta t} = \mathbf{A}_{i\Delta t} + \mathcal{N}_\theta(\mathbf{A}_{i\Delta t})\Delta t. \tag{33}$$

For global low-resolution simulations at $1/2°$ grid spacing, we set the time step $\Delta t = 24$ hours and integrate over a total period of $t = 4$ days, resulting in 4 discrete steps. This coarser temporal discretization is sufficient given the relatively slower dynamics captured at this scale.

In contrast, for regional high-resolution simulations at $1/12°$ grid spacing, the rapid evolution of oceanic processes—such as mesoscale eddies and submesoscale fronts—necessitates a finer time step to maintain numerical stability and accuracy. Accordingly, we adopt $\Delta t = 1800$ seconds (0.5 hours) and integrate over the same total period of $t = 4$ days, amounting to 192 small steps. However, to balance computational efficiency with fidelity, the neural network corrections $\mathbf{u}_\theta$ and $\dot{S}_\theta$ are applied only every 48 small steps (corresponding to one day).

This hybrid approach is formalized as follows: for each daily interval indexed by $k = 0, 1, 2, 3$ (spanning the 4-day period), we perform 48 sub-steps indexed by $j = 0, \ldots, 47$, with the full updates applied at the end of each daily block. For $j = 0$ to 47:

$$
\begin{aligned}
C_{(k\cdot48+j+1)\Delta t} = C_{(k\cdot48+j)\Delta t} \\
+ \Big[ -\mathbf{u}_{(k\cdot48+j)\Delta t}\nabla C_{(k\cdot48+j)\Delta t}\Big]\Delta t.
\end{aligned}
\tag{34}
$$

Then, at the end of the daily block:

$$
\begin{aligned}
\mathbf{u}_{(k\cdot48+48)\Delta t} &\leftarrow \mathbf{u}_{(k\cdot48+48)\Delta t} \\
&+ \mathbf{u}_\theta(\mathbf{u}_{(k\cdot48+48)\Delta t}, C_{(k\cdot48+48)\Delta t})
\end{aligned}
\tag{35}
$$

$$
C_{(k\cdot48+48)\Delta t} \leftarrow C_{(k\cdot48+48)\Delta t} + \dot{S}_\theta
\tag{36}
$$

This design ensures that the computationally intensive neural network evaluations are invoked daily, while the advection term is resolved at high temporal frequency to capture fast-changing dynamics.

Furthermore, as stated in the main text, our lightweight dynamical kernel imposes no substantial computational overhead on the simulations. As demonstrated in Table 8, in regional high-resolution simulations, incorporating the dynamical core results in only a 14% increase in inference time for the Ocean-E2E model.

Table 8: Inference time for Ocean-E2E 30-day forecast (batch size=1, on a single NVIDIA RTX 3090 GPU).

| Model Variant | Inference Time (ms) |
|---|---|
| Ocean-E2E w/ Dynamical Kernel | 195.29 |
| Ocean-E2E w/o Dynamical Kernel | 171.58 |

# D   DETAILS OF OUR NEURAL NETWORKS

## D.1   MULTI-SCALE CONVOLUTION

Inspired by Wu et al. (2024b), this section details the architectures of the neural network modules $\mathcal{M}_\theta$ (for modelling ocean surface geostrophic velocity). As shown in Figure 4, both modules leverage the `GABlock` (Group Attention Block) as their core building unit, designed to efficiently capture multi-scale spatio-temporal features.

The foundational `GABlock` (specifically, a `GASubBlock`) processes an input feature $\mathbf{X} \in \mathbb{R}^{B \times C_{in} \times H \times W}$. Initially, $\mathbf{X}$ is normalized to $\mathbf{X}_{\mathrm{norm1}} = \mathrm{BN}(\mathbf{X})$. This normalized feature is then passed through a Spatial Attention (`SA`) module. The `SA` module comprises an initial 1x1 convolutional projection $\mathcal{P}_1$, an activation function $\sigma_{\mathrm{act}}$ (e.g., GELU), a Spatial Gating Unit (`SGU`), and a final 1x1 convolutional projection $\mathcal{P}_2$. The core `SGU` employs depth-wise convolutions (`DWConv`$_0$) and depth-wise dilated convolutions (`DWDConv`$_{\mathrm{spatial}}$). Its output is formed by a feature transformation followed by a gating mechanism:

$$
\mathbf{F}_g = \mathrm{Conv}_{1\times1}\left(\mathrm{DWDConv}_{\mathrm{spatial}}\left(\mathrm{DWConv}_0\left(\sigma_{\mathrm{act}}\left(\mathcal{P}_1\left(\mathbf{X}_{\mathrm{norm1}}\right)\right)\right)\right)\right)
\tag{37}
$$

$$
\mathbf{F}_x, \mathbf{G}_x = \mathrm{split}\left(\mathbf{F}_g, \dim = 1\right)
\tag{38}
$$

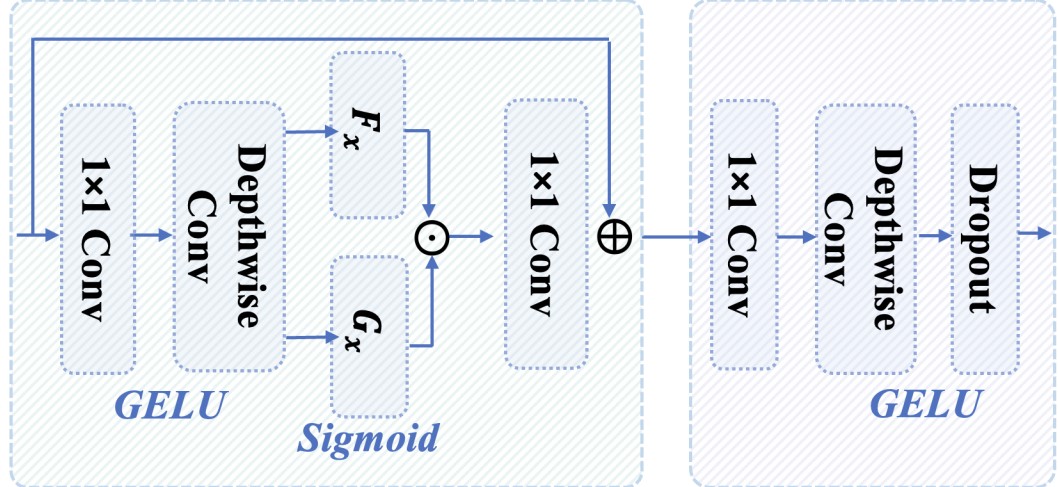

Figure 4: **Core components of the `GABlock`.** (Top) Core computational flow of the Multilayer Perceptron (Mlp) module, including a 1x1 convolution, a depthwise convolution, GELU activation, and Dropout. (Bottom) Core computational flow of the Spatial Attention (SA) module, illustrating the generation of feature $F_x$ and gating signal $G_x$ through an initial 1x1 convolution, GELU activation, and a depthwise convolution. The gating signal $G_x$ is then passed through a Sigmoid activation and element-wise multiplied with $F_x$. The result is processed by a final 1x1 convolution and integrated with the input via a residual connection. These two components collectively form the key information processing units within the GABlock.

$$\text{SGU}_{\text{out}} = \sigma_{\text{sigmoid}}(\mathbf{G}_x) \odot \mathbf{F}_x \tag{39}$$

The result from the `SA` module, $\mathcal{P}_2(\text{SGU}_{\text{out}})$, is integrated back into the input via a residual connection, scaled by a learnable parameter $\lambda_1$: $\mathbf{X}_{\text{attn}} = \mathbf{X} + \lambda_1 \cdot \text{DropPath}(\mathcal{P}_2(\text{SGU}_{\text{out}}))$. Subsequently, $\mathbf{X}_{\text{attn}}$ undergoes another normalization to $\mathbf{X}_{\text{norm2}} = \text{BN}(\mathbf{X}_{\text{attn}})$ and is processed by a Multilayer Perceptron (`Mlp`) module. The `Mlp` typically consists of two 1x1 convolutions sandwiching a depth-wise convolution, along with activations and dropout. Its output, $\text{Mlp}(\mathbf{X}_{\text{norm2}})$, is also added residually with a learnable scale $\lambda_2$: $\mathbf{X}_{\text{out}} = \mathbf{X}_{\text{attn}} + \lambda_2 \cdot \text{DropPath}(\text{Mlp}(\mathbf{X}_{\text{norm2}}))$.

The module for approximating the geostrophic velocity, $\mathbf{u}_g$, adopts a U-Net-like encoder-decoder architecture. It takes the initial conditions of geostrophic currents $\mathbf{u}_g \in \mathbb{R}^{B \times 2 \times H \times W}$ as input. The encoder path, composed of $L_e$ stages, utilizes downsampling convolutions (e.g., `ConvSC`) and `GABlocks` to extract multi-scale features, denoted as $\mathcal{H}_l$ for the $l$-th stage. Deeper encoder layers, particularly the bottleneck which processes $\mathcal{H}_{L_e}$ with stacked `GABlocks`, are crucial for capturing non-local spatial dependencies inherent in subgrid effects. The decoder path, with $L_d$ stages, symmetrically reconstructs spatial resolution using upsampling convolutions and `GABlocks`, integrating features from corresponding encoder stages $\mathcal{H}_{L_e-l+1}$ via skip connections. A final convolutional layer then maps the decoder's output to the future geostrophic velocity field $\mathbf{u}_g \in \mathbb{R}^{B \times 2 \times H \times W}$.

## D.2 NEURAL DATA ASSIMILATION NETWORK

Inspired by Liu et al. (2024), we apply an edge-reparameterization and attention-guided network to project the observation to analysis data, which achieves effcient assimilation for real-time application. It primarily consists of four parts, i.e., a convolutional layer (ConvL), a Kirsch-guided reparameterization module (KRM), a channel attention module (CAM), and a spatial attention module (SAM).

### D.2.1 CONVOLUTIONAL LAYER

The convolutional layer (ConvL) can defined as:

$$\text{ConvL}(x_{in}) = \mathcal{PR}(\mathcal{LN}(Conv(x_{in}))), \tag{40}$$

where $x_{in} \in \mathbb{R}^{C \times H \times W}$ is the input of ConvL. Conv, $\mathcal{LN}$, and $\mathcal{PR}$ represent the convolution operation, the layer normalization, and parametric rectified linear unit (PReLU), respectively.

### D.2.2  ATTENTION MECHANISM

The attention mechanism can realize the efficient allocation of information processing resources and can give more attention to key area of observatiuon while temporarily ignoring the missing locations. Therefore, the attention mechanism is used to focus on important information with high weights, ignore irrelevant information with low weights, and continuously adjust the weights during the network learning process. Therefore, a single model can extract more valuable feature information in different imaging environments. In this work, both channel attention and spatial attention are jointly exploited to further improve the assimilation performance.

**Channel Attention Module.** The channel attention mechanism is thus exploited to reconstruct the relationship between target analysis features and the input of different observation fields. The channel attention module (CAM) can be defined as:

$$\text{CAM} = \sigma(\text{MLP}(\text{Avg}(x_{in}^c)) + \text{MLP}(\text{Max}(x_{in}^c))), \tag{41}$$

where $x_{in}^c$ is the input of CAM, MLP denotes the multilayer perceptron, and $\sigma$ is the Sigmoid nonlinear activation function.

**Spatial Attention Module.** In data assimilation, the spatial attention module (SAM) is used to concentrate on crucial regions, which can be defined as:

$$\text{SAM} = \sigma\left(\text{Conv}^{7 \times 7}([\text{Avg}(x_{in}^s); \text{Max}(x_{in}^s)])\right), \tag{42}$$

where $x_{in}^s$ is the input of SAM, $[\,;\,]$ is exploited to concatenate two types of pooled features.

### D.2.3  KIRSCH-GUIDED REPARAMETERIZATION MODULE

To facilitate efficient assimilation in real-world MHWs forecasting, we employ the Kirsch-guided reparameterization module (KRM) with shared parameters, as proposed by Liu et al. Liu et al. (2024). The KRM is composed of three principal elements: (1) a standard convolutional layer that captures local spatial features; (2) an expand–squeeze convolutional layer that adaptively adjusts the receptive field; and (3) a family of eight directional edge-detection operators that guide the learning and inference of the convolutional kernels. Specifically, the standard convolutional and expand–squeeze operations can be formulated as follows:

$$F_n = W_n * x_{in}^k + B_n, \tag{43}$$

$$F_{es} = W_s * \left(W_e * x_{in}^k + B_e\right) + B_s, \tag{44}$$

where $x_{in}^k$ denotes the input of KRM. $W_n$, $W_s$, $W_e$, $B_n$, $B_s$, $B_e$ are the weights and bias of convolution operation. $F_n$ represents the output of the normal convolutional layer. $F_{es}$ denotes the output of the expanding-and-squeezing convolutional layer. We first reparameterize the Eqs. (43) and (44). Subsequently, we merge them into a single normal convolution with parameters $W_{es}$ and $B_{es}$:

$$W_{es} = \text{perm}\left(W_e\right) * W_s, \tag{45}$$

$$B_{es} = W_s * \text{rep}\left(B_e\right) + B_s, \tag{46}$$

where $\text{perm}(\cdot)$ denotes the permute operation which exchanges the 1st and 2nd dimensions of a tensor, $\text{rep}(\cdot)$ denotes the spatial broadcasting operation. And the predefined eight-direction Kirsch edge filters $K_i$ are incorporated into the reparameterization module. To memorize the edge features, the input feature $x_{in}^k$ will first be processed by $C \times C \times 1 \times 1$ convolution and then use a custom Kirsch filter to extract the feature map gradients in eight different directions. Therefore, the edge information in eight directions can be expressed as follows:

$$\begin{aligned} F_K^i &= \left(S_K^i \odot K_i\right) \otimes \left(W_i * x_{in}^k + B_i\right) + B_{K_i} \\ &= W_K^i \otimes \left(W_i * x_{in}^k + B_i\right) + B_{K_i}, \end{aligned} \tag{47}$$

where $W_i$ and $B_i$ are the weights and bias of $1 \times 1$ convolution for branches in eight directions, $S_K^i$ and $B_{K_i}$ are the scaling parameters and bias with the shape of $C \times 1 \times 1 \times 1$, $\odot$ indicates the

channel-wise broadcasting multiplication. The combined edge information, extracted by the scaled Kirsch filters, is given by:

$$F_{\text{K}} = \sum_{i=1}^{8} F_K^i. \tag{48}$$

Therefore, the final weights and bias after reparameterization can be expressed as:

$$W_{\text{rep}} = W_n + W_{es} + \sum_{i=1}^{8} \left( \text{perm} \left( W_i \right) * W_{\text{K}}^i \right), \tag{49}$$

$$B_{\text{rep}} = B_n + B_{es} + \sum_{i=1}^{8} \left( \text{perm} \left( B_i \right) * B_{\text{K}_i} \right). \tag{50}$$

Finally, the output can be obtained using a single normal convolution in the inference stage:

$$F = W_{\text{rep}} * x_{in}^k + B_{\text{rep}}. \tag{51}$$

### D.2.4 THE ASSIMILATION NETWROK

We fist define the basic block:

$$res_1 = \mathcal{PR}(\mathcal{G}(Conv(x_{in}^b))), \tag{52}$$

$$res_2 = \mathcal{PR}(\mathcal{G}(Conv(res_1))), \tag{53}$$

$$res_{attention} = SAM(CAM(res2)), \tag{54}$$

$$res_3 = \mathcal{PR}(\mathcal{G}(Conv(res_{attention})) + x_{in}^b), \tag{55}$$

$$res_4 = \mathcal{PR}(\mathcal{G}(Conv(res_3))), \tag{56}$$

where, $x_{in}^b$ is the input of basic block, $\mathcal{G}$ represents GroupNorm. In summary, the basic block can be defined as:

$$res_4 = Block(x_{in}^b). \tag{57}$$

Therefore, for the obsevation data $x_o$, it can be processed to analysis data as follow:

$$y_a^1 = Block(KRM(x_o) + x_o), \tag{58}$$

$$y_a^2 = Block(KRM(y_a^1) + y_a^1), \tag{59}$$

$$y_a^3 = Block(KRM(y_a^2) + y_a^2), \tag{60}$$

$$y_a^4 = Block(KRM(y_a^3) + y_a^3), \tag{61}$$

$$y_a^5 = Block(KRM(y_a^4) + y_a^4), \tag{62}$$

$$y_a^{out} = \mathcal{C}(y_a^5), \tag{63}$$

where, $\mathcal{C}$ denotes the final convolution operation, which maps the latent information to the analysis data.

## E EXPERIMENTS DETAILS

### E.1 EVALUATION METRIC

**Definition of MHWs**. The Hobday et al. (2016) definition of MHW categories is applied to detect MHWs from original SST field. More precisely, any interval in which the SST anomalies surpass the 90th percentile for a minimum of five days in a row is classified as an MHW event.

**Metrics**. In this study, to evaluate the performance of our model, we use three metrics: Root Mean Square Error (RMSE), Critical Success Index (CSI) and Bias (BIAS). RMSE represents the overall performance of our SST anomaly forecast, which is calculated as:

$$\text{RMSE}(t) = \sqrt{\frac{1}{H \times W} \sum_{i,j} \omega_{i,j} (C_{i,j}(t) - \tilde{C}_{i,j}(t))^2} \tag{64}$$

Where $C_{i,j}(t)$ and $\tilde{C}_{i,j}(t)$ is the ground truth and prediction of SST anomaly at lead time $t$, $\omega_{i,j}$ is the weighting coefficient and the subscript $(i,j)$ indicates the data at grid point $(i,j)$ while $H$ and $W$ follow the same definition in Section 2. CSI is the metric that evaluate the forecast ability of extreme events and can be expressed as:

$$\text{CSI}(t) = \frac{\text{TP}}{\text{TP} + \text{FP} + \text{FN}} \tag{65}$$

Where TP (True Positive) denotes the number of cases that a MHW event is accurately predicted. FP (False Positive), FN (False Negative), and TN (True Negative) follow a similar definition.

BIAS is used to evaluate the systematic difference between the analysis field and the ground truth:

$$\text{BIAS}(t) = \frac{1}{H \times W} \sum_{i,j} \omega_{i,j}(C_{i,j}(t) - \tilde{C}_{i,j}(t)) \tag{66}$$

## E.2 MODEL TRAINING

All baseline models and Ocean-E2E were trained under identical experimental configurations to ensure fair comparison. The training process employed 100 epochs with an initial learning rate of $1 \times 10^{-3}$, utilizing a step-wise learning rate scheduler to dynamically adjust training parameters until convergence was achieved. For objective evaluation, model checkpoints demonstrating optimal performance on the validation set were selected as final candidates for comparative analysis.

## F RELATED WORK AND LIMITATIONS

### F.1 NUMERICAL MODEL BASED OCEAN FORECASTING

Traditional MHWs forecasting predominantly relies on numerical models solving oceanic primitive equations. These approaches are typically categorized into two types: (1) seasonal forecasting, which predicts large-scale MHW characteristics including onset timing, duration, and intensity evolution Jacox et al. (2022); Brodie et al. (2023), and (2) sub-seasonal forecasting, which resolves finer spatiotemporal variations at the cost of reduced prediction horizons Benthuysen et al. (2021); Yu et al. (2024). While physics-based models demonstrate reasonable forecast skill, they face two principal limitations: prohibitive computational costs scaling with model resolution Xiong et al. (2023); Wang et al. (2024), and systematic underestimation of extreme MHW magnitudes Jacox et al. (2022).

### F.2 DEEP LEARNING BASED OCEAN FORECASTING

Machine learning (ML) techniques have emerged as promising alternatives for oceanic modeling, offering orders-of-magnitude speedup over conventional methods Liu & Ma (2024); Hao et al. (2025). Recent efforts have developed spatio-temporal architectures including NMO Wu et al. (2024a), neXtSIM Finn et al. (2024), and OceanVP Shi et al. (2024b) for regional ocean modeling, alongside specialized models for specific oceanic phenomena such as ENSO Ham et al. (2019); Chen et al. (2025), MJO Kim et al. (2021); Delaunay & Christensen (2022); Shin et al. (2024), and MHWs Shi et al. (2024a); Jacox et al. (2022); Shu et al. (2025). Building on advances in atmospheric foundation models Pathak et al. (2022); Bi et al. (2023), the community has recently proposed global ocean foundation models like AI-GOMS Xiong et al. (2023), Xihe Wang et al. (2024), and WenHai Cui et al. (2025).

### F.3 DEEP LEARNING BASED DATA ASSIMILATION

Deep learning (DL) has shown transformative potential in weather and climate data assimilation (DA), with progress evolving along three directions: (1) Conceptual validation using idealized systems like Lorenz models and shallow water equations Brajard et al. (2020); Arcucci et al. (2021); Fablet et al. (2021); Legler & Janjić (2022); (2) Component replacement within conventional DA frameworks, including physical parameterization Hatfield et al. (2021), observation operators Liang et al. (2022); Stegmann et al. (2022), and optimization cost functions Melinc & Zaplotnik (2024); Xiao et al. (2024); (3) End-to-end DA system replacement, exemplified by FengWu-Adas and DiffDA that

implement complete DL-based assimilation pipelines using ERA5 reanalysis data Chen et al. (2023b); Huang et al. (2024).

### F.4 LIMITATIONS

Despite the promising results demonstrated by our proposed Ocean-E2E framework, several limitations should be acknowledged. First, the hybrid approach relies on certain simplifications of the underlying physical processes, which may not fully capture the complex, multi-scale interactions in certain oceanic regimes. Second, the current model does not explicitly account for potential feedback mechanisms between marine heatwaves and larger-scale climate modes, which may affect the accuracy of long-term forecasts. Future work could focus on refining the physical constraints, expanding the evaluation across diverse marine environments, and exploring more efficient architectures to enhance robustness and operational feasibility.

## G ADDITIONAL RESULTS

### G.1 ADDITIONAL RESULTS OF OUR END-TO-END FORECAST SYSTEM

In this section, we are going to show additional results of our end-to-end forecast system.

**Details of the performance of our assimilation framework**. Figure 5 illustrates the performance of our assimilation, including a snapshot of our analysis field and temporal evolution of our assimilation error.

**Additional comparsions between our framework and S2S**. To be specific, we take the S2S analysis field as the groundtruth, and use it to initialize our Ocean-E2E. The atmospheric boundary condition is acquired through one of the state-of-the-art AI-driven weather forecast model: OneForecast Gao et al. (2025). As shown in Table 10, our model still performs better than S2S.

**Ablation Analysis on Error Propagation**. We further present the ablation results for the core components of the marine heatwave forecasting task to investigate how errors propagate through the atmospheric forcing forecast module $\mathcal{N}_\theta$ and the assimilation module $\phi_\theta^a$. As shown in Table 9, comparing the full Ocean-E2E model with variants using real atmospheric forcing (w/o $\mathcal{N}_\theta$) and real initial conditions (w/o $\phi_\theta^a$), we observe that the error in the atmospheric forecast module accumulates gradually over time (e.g., the gap between w/o $\mathcal{N}_\theta$ and Full model widens at 40-day lead time). This underscores the critical role of accurate atmospheric forcing fields. In contrast, the assimilation module does not exhibit a significant cumulative error effect, as it primarily impacts the state fields in the immediate vicinity of the assimilation timestamp, thereby exerting a localized effect.

Table 9: Ablation study on core components regarding error propagation. We compare the RMSE of Global SSTA across different lead times (20 and 40 days) for the years 2020 and 2021.

| Model Variants | 2020 | | 2021 | |
|---|---|---|---|---|
| | 20-day | 40-day | 20-day | 40-day |
| w/o $\mathcal{N}_\theta$ (real atmosphere) | 0.5944 | 0.8325 | 0.7297 | 0.8132 |
| w/o $\phi_\theta^a$ (real IC) | 0.5832 | 0.8691 | 0.6965 | 0.8531 |
| **Ocean-E2E (Full)** | **0.6047** | **0.8747** | **0.7447** | **0.8729** |

### G.2 ADDITONAL EVALUATIONS OF OUR MODEL'S PERFORMANCE

In this section, we present the power spectrum of the squared modulus of the SSTA gradient, $|\nabla C|^2$, on day 20. As shown in Figure 6, our Ocean-E2E model, especially at small scales (corresponding to a large wave number), is closer to the ground truth. This additional evaluation metric demonstrates the accuracy of our model in simulating the heatwave system.

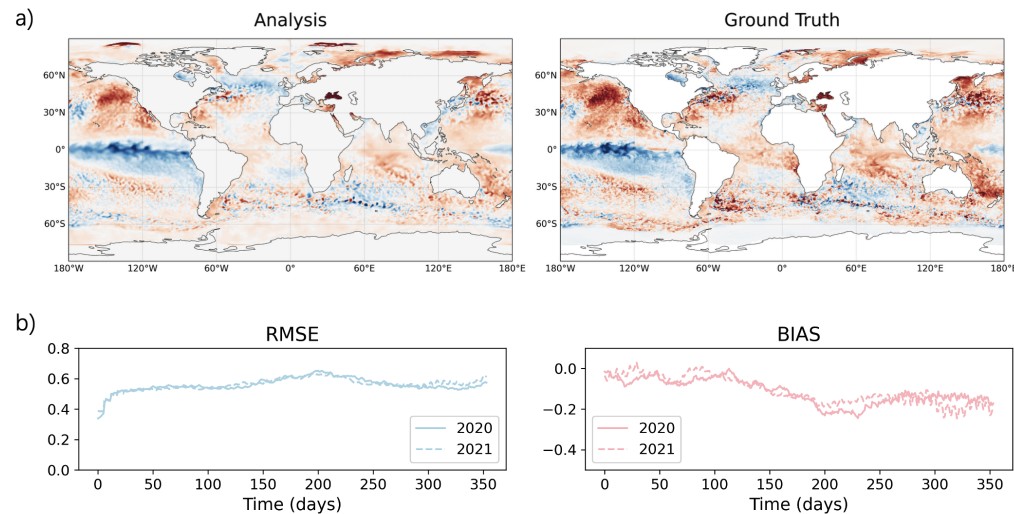

Figure 5: Results of our end-to-end assimilation framework. a) Snapshots of our analysis field and the groundtruth after 300 days of the initialization of our assimilation in 2020 b) Temporal evolution of RMSE and BIAS during the assimilation period.

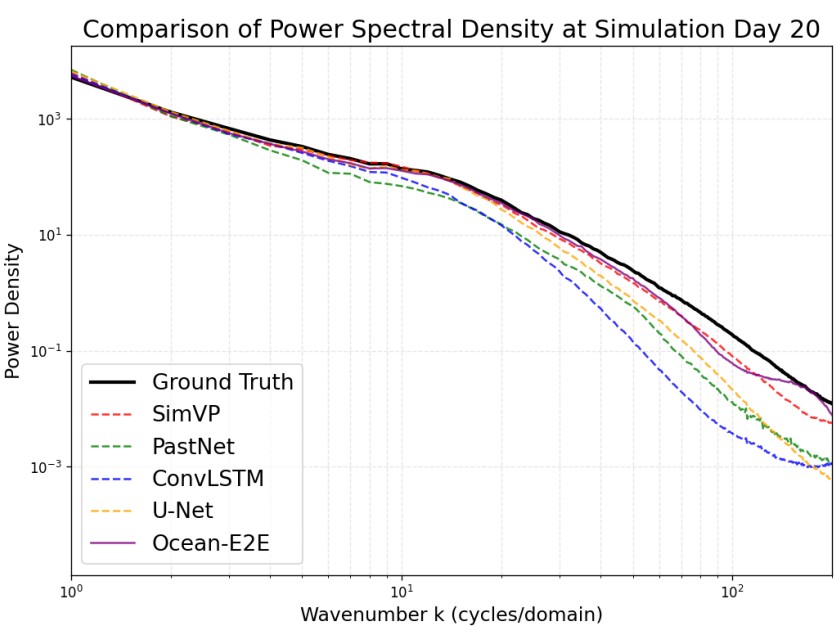

Figure 6: Power Spectrum of $|\nabla C|^2$ on day 20 from our regional simulation experiment.

### G.3 VISUALIZATION OF GLOBAL MHWS SIMULATION

In this section, we visualize the global MHWs forecast result of our Ocean-E2E, as shown in Figure 7-9. **We plot the global SSTA field, where darkred parts indicate strong MHWs.**

### G.4 VISUALIZATION OF GLOBAL MHWS FORECAST

In this section, we visualize the global MHWs forecast result of our Ocean-E2E, as shown in Figure 10-12. **We plot the global SSTA field, where darkred parts indicate strong MHWs.**

Table 10: Additional comparison with S2S. The average results for global SSTA of RMSE are recorded. A small RMSE (↓) indicates better performance. The best results are in **bold**, and the second best are with underline.

| MODEL | METRIC (RMSE) | | | |
|---|---|---|---|---|
| | TIME INTERVAL | | | |
| | 14-DAY | 21-DAY | 28-DAY | 35-DAY |
| S2S (2020) | 0.5249 | 0.5994 | 0.6618 | 0.7086 |
| OCEAN-E2E (2020) | **0.4321** | **0.5242** | **0.5891** | **0.6304** |
| OCEAN-E2E (PROMOTION) | 17.8 % | 12.5 % | 11.0 % | 11.0 % |
| S2S (2021) | 0.5175 | 0.5703 | 0.6121 | 0.6448 |
| OCEAN-E2E (2021) | **0.4350** | **0.5167** | **0.5626** | **0.6022** |
| OCEAN-E2E (PROMOTION) | 15.9 % | 9.4 % | 8.1 % | 6.6 % |

## G.5 VISUALIZATION OF REGIONAL HIGH RESOLUTION SIMULATION

In this section, we visualize the regional (western Atlantic) simulation, as shown in Figure 13-15. **We plot the regional SSTA field, where yellow parts indicate MHWs.**

# H LARGE LANGUAGE MODEL (LLM) USE DISCLOSURE

No large language models were used in the creation of this work for the purposes of writing assistance, literature retrieval, research ideation, or any other aspect of the research and manuscript preparation process.

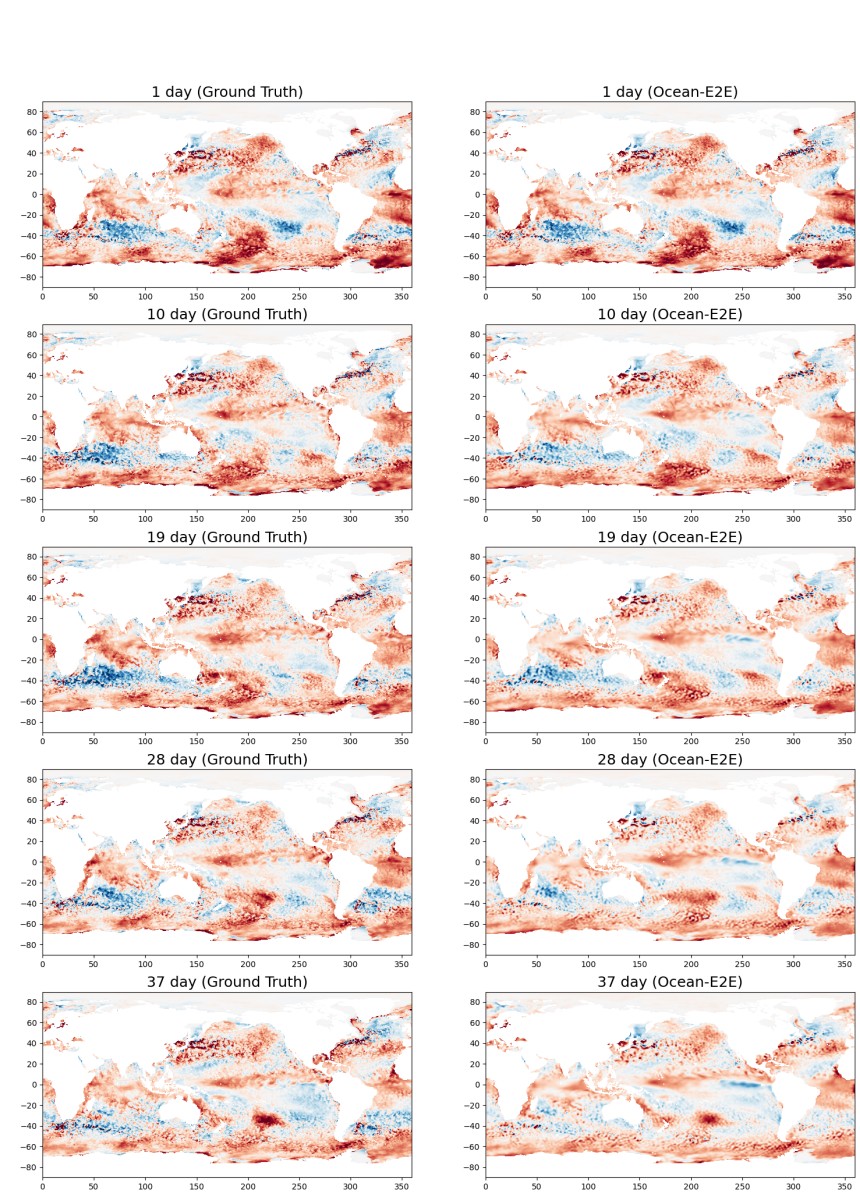

Figure 7: Global Simulation of MHWs Initialized on January 13, 2020

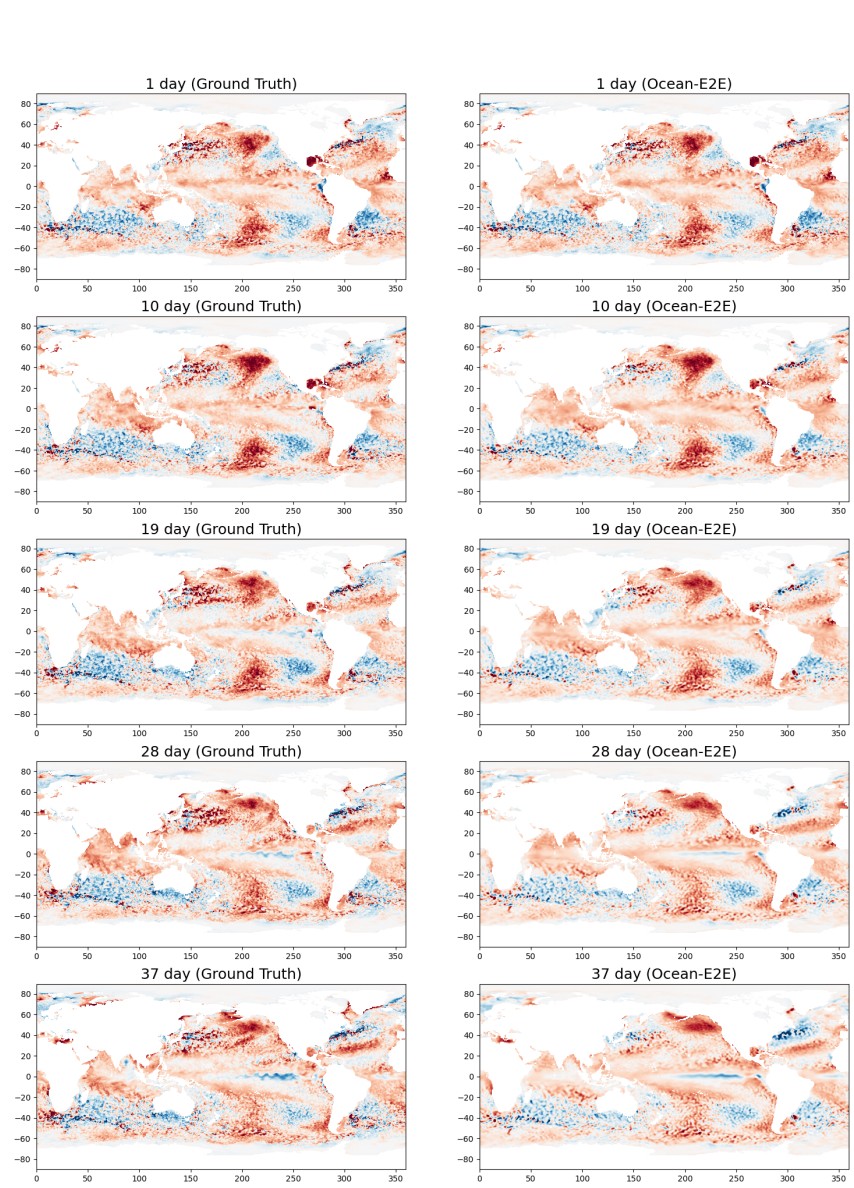

Figure 8: Global Simulation of MHWs Initialized on May 12, 2020

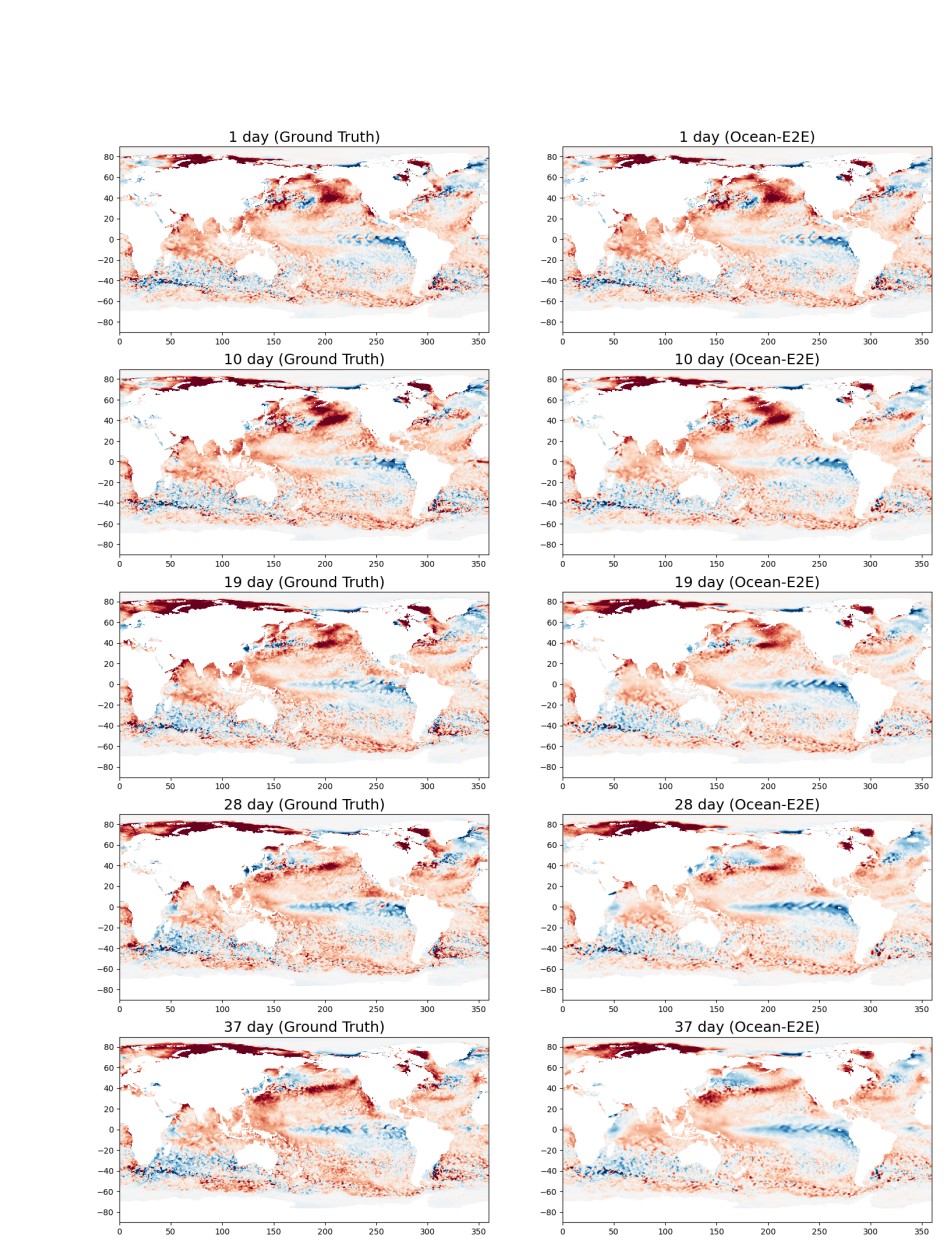

Figure 9: Global Simulation of MHWs Initialized on July 11, 2020

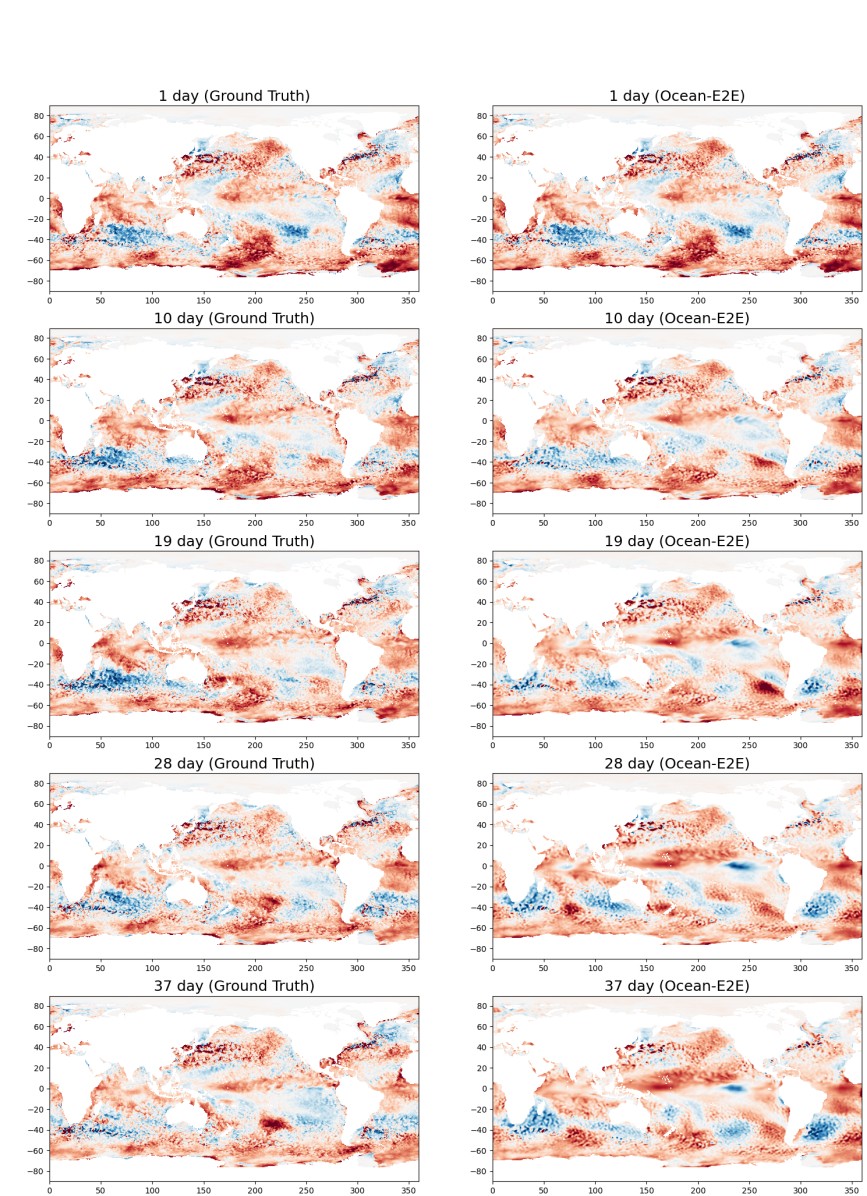

Figure 10: Global Forecast of MHWs Initialized on January 13, 2020

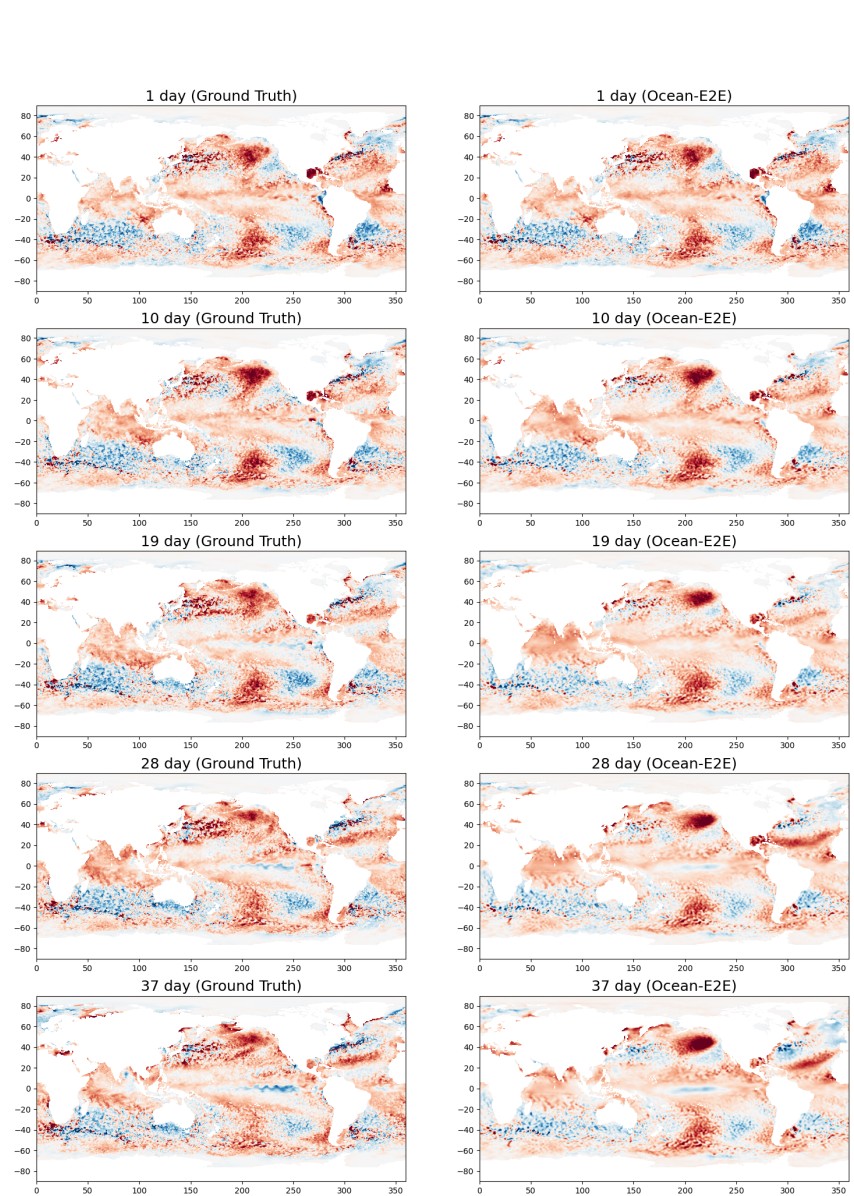

Figure 11: Global Forecast of MHWs Initialized on May 12, 2020

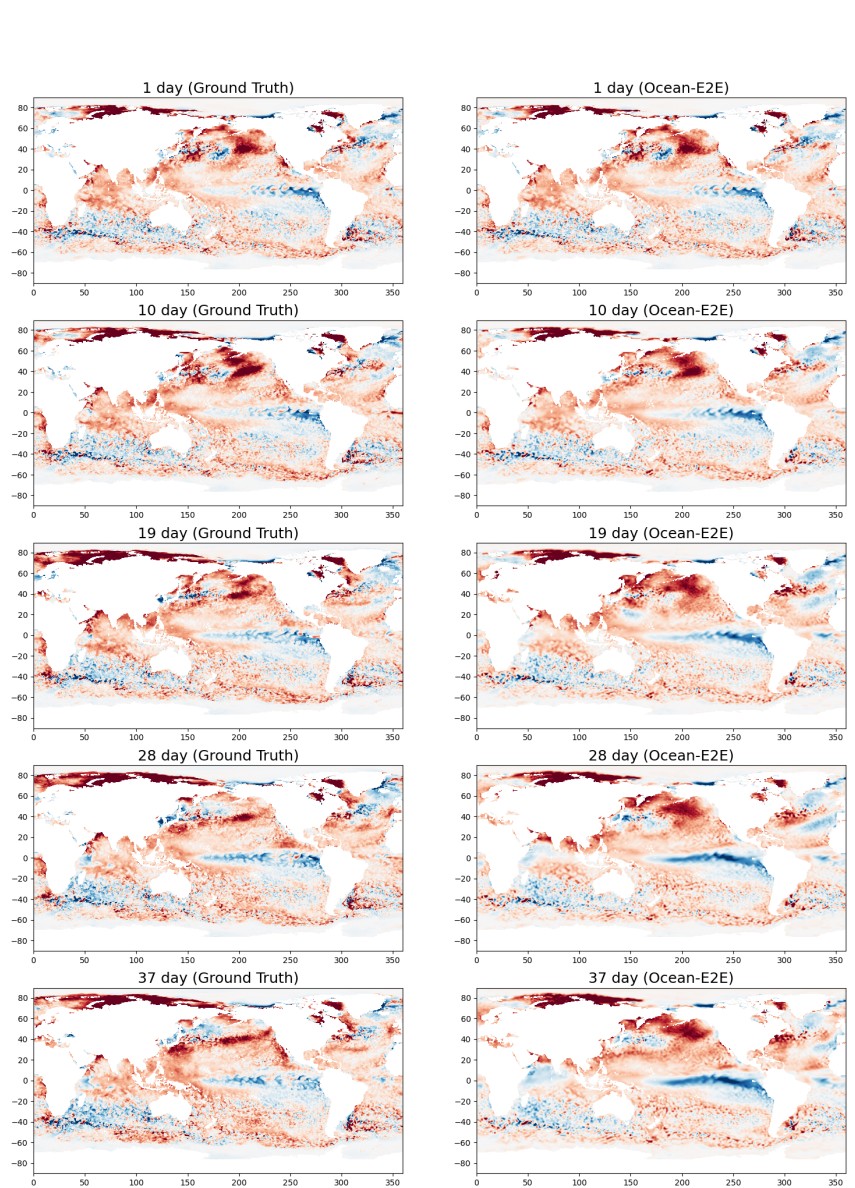

Figure 12: Global Forecast of MHWs Initialized on July 11, 2020

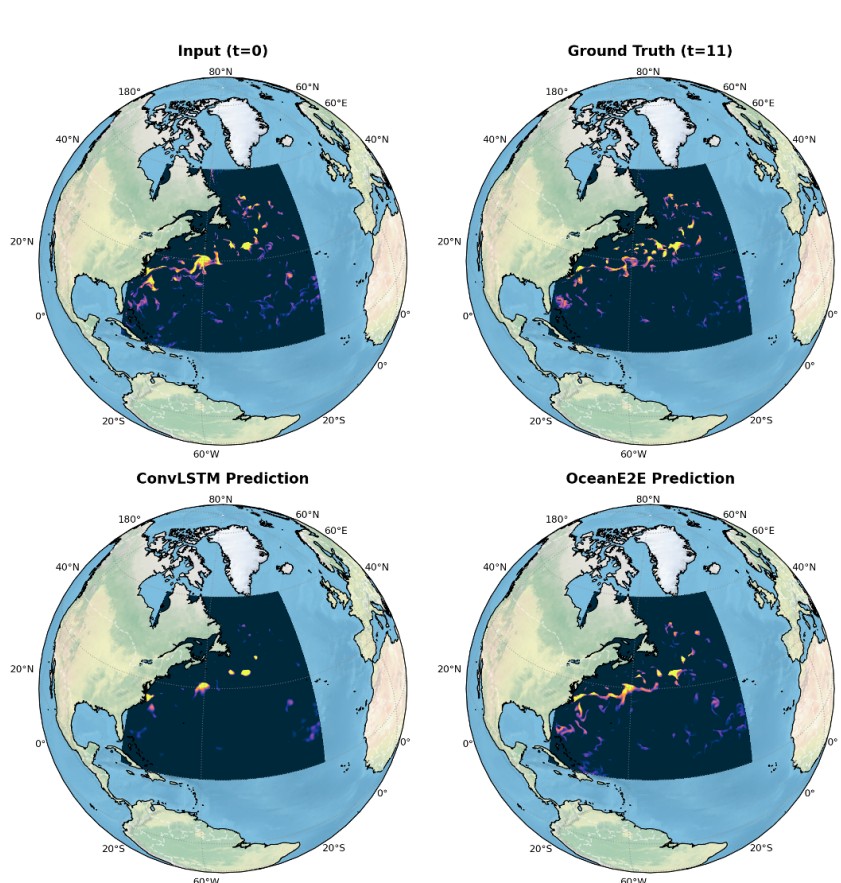

Figure 13: Regional Simulation of West Atlantic Initialized on January 13, 2020

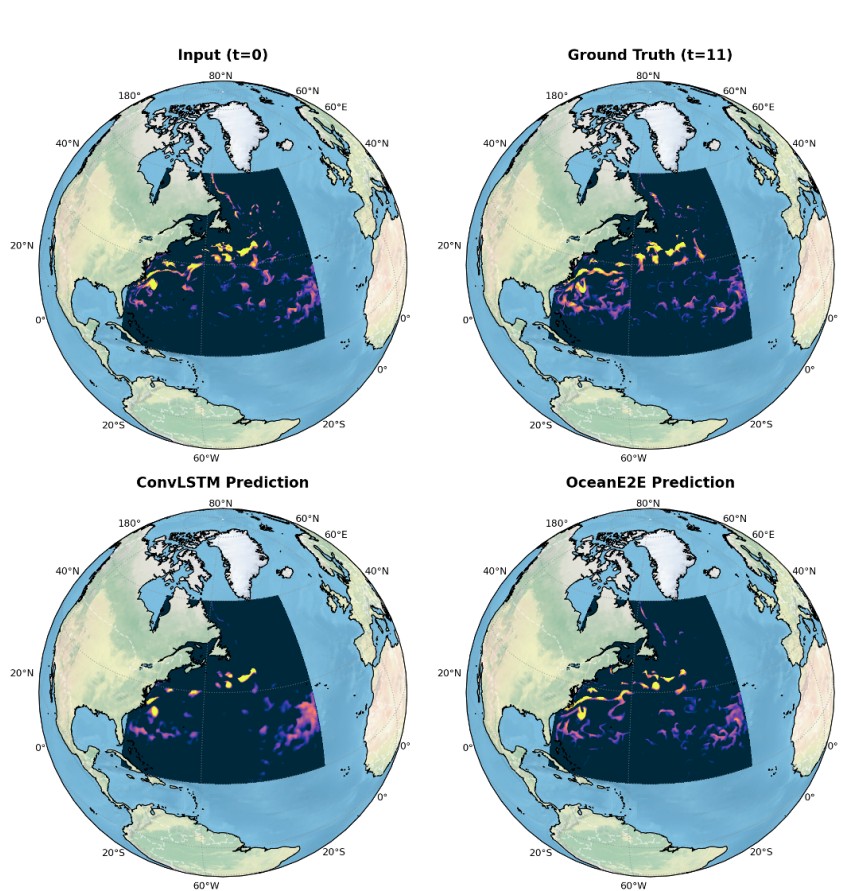

Figure 14: Regional Simulation of West Atlantic Initialized on March 13, 2020

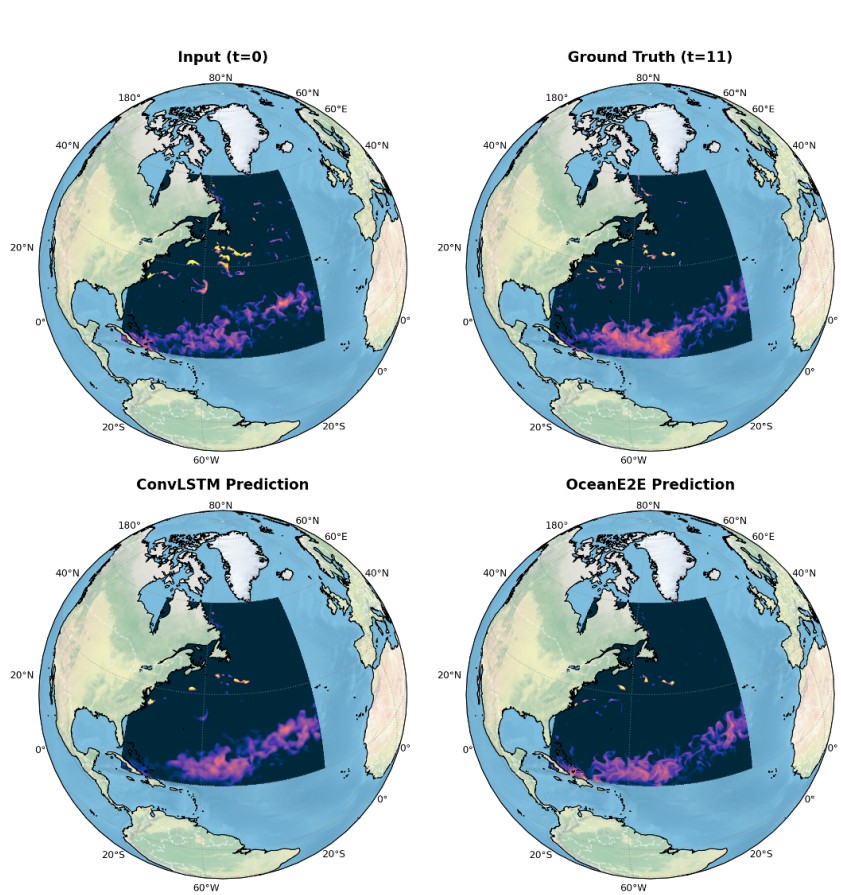

Figure 15: Regional Simulation of West Atlantic Initialized on May 12, 2020

