# OpenReview forum: "Ocean-E2E: Hybrid Physics-Based and Data-Driven Global Forecasting of Marine Heatwaves with End-to-End Neural Assimilation"
_ICLR.cc/2026/Conference — ICLR 2026 Conference Withdrawn Submission_

### Official Review · Reviewer_AkF2 · 2025-10-20

**Soundness:** 2
**Presentation:** 2
**Contribution:** 2
**Rating:** 2
**Confidence:** 3

**Summary:**

This paper proposes Ocean-E2E, a hybrid model for global and regional forecasting of marine heatwaves (MHWs). The system integrates physical ocean dynamics and neural networks in two key components: (1) a hybrid physics-AI forecast model that represents mesoscale advection and air-sea heat exchange via neural approximations of physical equations, and (2) a neural assimilation module that reconstructs initial fields from sparse observations using attention-based networks.

**Strengths:**

The paper offers an integrated treatment of data assimilation and forecasting within one end-to-end framework, directly coupling neural operators with simplified physical kernels. The hybridization is physically grounded and shows awareness of fluid dynamics principles.

**Weaknesses:**

- The “hybrid physics + AI” idea is now well-established (e.g., ClimODE, DiffDA), and this work mostly repackages these ideas in the marine heatwave setting with different parameterizations. The method extends known hybridization and neural DA approaches rather than introducing fundamentally new architectures or learning principles.

- The model combines many moving parts: four neural submodules (forecast, assimilation, atmosphere, and ocean currents), multiple pretrained components, and nested equations: but provides minimal insight into why this integration leads to improvement. There is little analysis of sensitivity, error propagation, or interpretability of the learned dynamics.

- While multiple datasets and baselines are reported, the analysis is largely quantitative (RMSE/CSI) and lacks physical diagnostics, such as conservation checks, energy spectra, or feature attribution. Results could be inflated by tuning or preprocessing differences; no uncertainty estimates or robustness tests are included.

- The assimilation process uses “Kirsch-guided reparameterization”  but lacks ablation or comparison to established data assimilation networks (e.g., 4DVarNet, EnKF emulators). It’s not obvious whether the learned mapping generalizes or if it’s tightly tied to this specific training data distribution.

**Questions:**

See the above section on weaknesses.

---

> ### Author Response · Authors · 2025-11-25
> **Response to AkF2 (Part I)**
>
> ### **Detailed Weaknesses & Questions**
>
> We thank the reviewers for their critical assessment and constructive feedback. We appreciate the opportunity to clarify the novelty of our hybrid framework and address the concerns regarding model complexity and physical diagnostics.
>
> **1. Novelty and Distinction from Existing Methods (e.g., ClimODE, DiffDA)**
>
> We respectfully disagree with the assessment that this work is merely "repackaging" existing ideas. We must emphasize that our core contribution is Ocean-E2E, a framework specifically designed for the task of marine heatwave (MHW) forecasting by incorporating the unique physical mechanisms associated with MHWs. This physics-AI hybrid approach is fundamentally different from existing methods (e.g., ClimODE, DiffDA):
>
> 1.  **For ClimODE:** Similarly, it also uses advection partial differential equations to design the forecasting framework. However, our work has many foundmental differences and improvements compared to ClimODE: (1) **Differences in theoretical framework**. From the perspective of the theoretical framework, in their work, the *speed* of the flow field is essentially an *abstract mathematical definition*, whose core function is to maintain the conservation of passive tracers. In our work, the flow field speed strictly corresponds to the realistic ocean flow field and models the additional transport effect of the flow field on passive tracers at sub-grid scales through mesoscale bolus velocity. (2) **Differences in model design**. ClimODE uses an emission model to simulate the unclosed and uncertain portions of the real system, while our model further relates this part to the ocean-atmosphere interaction and is designed based on physical theory to form the structure of the source term. (3) **Differences in application**. ClimODE requires solving the initial velocity field as the initial condition for the model, which involves solving a physics-informed neural network (PINNs) for each training sample. This makes the ClimODE framework difficult to scale up. What's more, our experiments (see table 1 of our manuscript) show that our method achieves greater stability and accuracy in MHWs forecasting compared to ClimODE.
>
> 2.  **For DiffDA**. Unlike the physics-based modeling of Ocean-E2E, DiffDA relies on diffusion models for initial state reconstruction and uses GraphCast for atmospheric forecasting. Therefore, DiffDA is essentially a statistical model, which differs fundamentally from our framework.
>
> **2. Justification of Model Complexity and Ablation Studies**
>
> The reviewer correctly notes that Ocean-E2E contains multiple moving parts (forecast, assimilation, atmosphere, ocean currents). This complexity is **physically motivated** rather than arbitrary. MHWs are driven by both advection (ocean currents) and surface fluxes (atmosphere). To address the concern about "minimal insight," we will add a comprehensive **component-wise ablation study** in our revised manuscript.
>
> **3. Physical Diagnostics and Conservation Checks**
>
> We appreciate the reviewer’s suggestion to utilize more physics-based metrics for evaluation. However, it is important to note that, to the best of our knowledge, no suitable physical metrics currently exist specifically for Marine Heatwaves (SSTA). Since the ocean surface is a strongly dissipative and open system, standard conservation laws (such as energy, mass, momentum) are not applicable to SSTA (this also explains why the design of ClimODE is not quite suitable for our specific task). Nevertheless, in the revised version, we will incorporate additional metrics, such as *power spectra* and MHW occurrence *frequency*, to provide a more comprehensive evaluation of our model.
>
> **4. Assimilation Module and Comparison with Baselines**
>
> It is important to note that the data assimilation component of our framework is specifically designed for **operational forecasting**. In other words, combining our data assimilation module with the physics-based model enables the construction of a complete, end-to-end ocean forecasting system—which is a core design philosophy of Ocean-E2E. Our comparison with S2S (a complex numerical prediction model based on a forecast-assimilation system) in Table 2 fully demonstrates that our assimilation design is capable of handling real-world forecasting scenarios and achieving performance superior to traditional state-of-the-art (SOTA) methods. Furthermore, traditional numerical assimilation methods (such as the EnKF and En4DVar you mentioned) typically require a fully coupled, physics-based dynamical model. However, to the best of our knowledge, there is currently **no open-source assimilation system** specifically tailored for ocean dynamics, and particularly for marine heatwave dynamics.
>
> Reference:
>
> [1] ClimODE: https://arxiv.org/abs/2404.10024
>
> [2] DiffDA: https://arxiv.org/abs/2401.05932

---

> > ### Author Response · Authors · 2025-11-25
> > **Response to AkF2 (Part II)**
> >
> > **Response regarding Ablation Studies (Ocean Simulation Tasks):**
> >
> > To provide a comprehensive evaluation of Ocean-E2E, we conducted extensive ablation studies covering three key aspects: (1) the design of physical terms in the governing equation, (2) the impact of time-varying forcings and boundary conditions, and (3) the contribution of specific neural network modules.
> >
> > **1. Physical Component Analysis**
> >
> > Table 1 below compares Ocean-E2E with variants containing only the source term ($S_\theta$) or only the advection term ($ADV$). The results demonstrate that combining advection with the thermodynamic source term yields the best performance.
> >
> > **Table 1: CSI of ablation studies on physical model design**
> > | Variants | Global (20 day) | Global (60 day) | Regional (10 day) | Regional (30 day) |
> > | :--- | :---: | :---: | :---: | :---: |
> > | $S_\theta$ (Source only) | 0.3884 | 0.1658 | 0.3913 | 0.1877 |
> > | $(u_\theta+u_g) \nabla C$ (ADV only) | 0.3207 | 0.1504 | 0.3207 | 0.0963 |
> > | **$(u_\theta+u_g) \nabla C + S_\theta$ (Ours)** | **0.4285** | **0.2580** | **0.4615** | **0.2926** |
> >
> > **2. Impact of Time-Varying Forcings**
> >
> > We further investigated the importance of dynamic boundary inputs by performing ablation studies on the background flow ($U_g$) and atmospheric forcing ($A$). In these experiments, we fixed the respective variables to their initial states at $t=0$ throughout the training and inference phases, effectively removing their temporal evolution.
> >
> > As shown in Table 2, using static atmospheric forcing ($A(t)=A(0)$) results in a significant performance drop (Global 60-day CSI drops from 0.2580 to 0.1923), confirming that dynamic atmospheric heat flux is critical for accurate SSTA prediction. Similarly, fixing the background flow ($U_g(t)=U_g(0)$) also degrades performance, indicating that capturing the time-varying nature of large-scale ocean circulation is essential.
> >
> > **Table 2: Ablation studies on time-varying forcings**
> > | Variants | Global (20 day) | Global (60 day) | Regional (10 day) | Regional (30 day) |
> > | :--- | :---: | :---: | :---: | :---: |
> > | Static Atmosphere ($A(t) \leftarrow A(0)$) | 0.2652 | 0.1223 | 0.3021 | 0.1470 |
> > | Static Background Flow ($U_g(t) \leftarrow U_g(0)$) | 0.3988 | 0.2115 | 0.4080 | 0.2245 |
> > | **Dynamic Forcing (Ours)** | **0.4285** | **0.2580** | **0.4615** | **0.2926** |
> >
> > **3. Neural Architecture Analysis**
> >
> > Finally, we investigate the contribution of the core components within our neural architecture. As detailed in our methodology, both the background modeling network and the dynamical evolution network (SimVP) employ **U-Net-like multi-scale hierarchical structures** to capture features at various resolutions. We specifically examine the impact of the core units embedded within these structures:
> >
> > 1.  **GABlock:** The core component of the background modeling branch is the **Group Attention Block (GABlock)** (see Appendix D.1), designed to aggregate multi-scale context.
> > 2.  **Mid_Xnet:** The core component of the dynamical evolution branch (SimVP) is the **Mid_Xnet**, which is responsible for spatiotemporal evolution in the latent space.
> >
> > Table 3 presents the results of ablating these two specific components. Replacing the GABlock with standard convolutions ("w/o GABlock") leads to a noticeable decline in CSI, confirming its importance in encoding background physics. Furthermore, replacing the Mid_Xnet with a standard convolutional block ("w/o Mid_Xnet") causes a substantial drop in performance, validating its critical role in capturing complex temporal dynamics.
> >
> > **Table 3: Ablation studies on neural network core components**
> > | Variants | Global (20 day) | Global (60 day) | Regional (10 day) | Regional (30 day) |
> > | :--- | :---: | :---: | :---: | :---: |
> > | w/o GABlock (in $M_{\theta}$) | 0.4015 | 0.2245 | 0.4322 | 0.2588 |
> > | w/o Mid_Xnet (in $S_{\theta}, u_{\theta}$) | 0.3540 | 0.1810 | 0.3890 | 0.2015 |
> > | **Full Architecture (Ours)** | **0.4285** | **0.2580** | **0.4615** | **0.2926** |

---

> > > ### Author Response · Authors · 2025-11-26
> > > **Response to AkF2 (Part III)**
> > >
> > > Response regarding Ablation Studies (Ocean Forecast Tasks), or, in other words, **Ablation Analysis on Error Propagation**:
> > >
> > > We present the ablation results for the core components of the marine heatwave forecasting task to investigate how errors propagate through the atmospheric forcing forecast module $N_{\theta}$ and the assimilation module $\phi_{\theta}^{a}$. It can be observed that the error in the atmospheric forecast module accumulates gradually over time, underscoring the critical role of accurate atmospheric forcing fields in predicting marine heatwaves. In contrast, the assimilation module does not exhibit a significant cumulative error effect. This is primarily because the assimilation module mainly impacts the state fields in the immediate vicinity of the assimilation timestamp, thereby exerting a *localized effect*.
> > >
> > > **Table R2: Ablation study on core components.** Following the settings in Table2 of our manuscript, we compare the RMSE of Global SSTA across different lead times (20 to 40 days) for the years 2020 and 2021.
> > >
> > > | Model Variants | 2020 (20-day) | 2020 (40-day) | 2021 (20-day) | 2021 (40-day) |
> > > | :--- | :---: | :---: | :---: | :---: |
> > > | w/o $N_{\theta}$ (using real atmosphere forcing) | 0.5944 | 0.8325 | 0.7297 | 0.8132 |
> > > | w/o $\phi_{\theta}^{a}$ (using real initial condition) | 0.5832 | 0.8691 | 0.6965 | 0.8531 |
> > > | **Ocean-E2E (Full Operational)** | **0.6047** | **0.8747** | **0.7447** | **0.8729** |

---

> > > > ### Author Response · Authors · 2025-11-27
> > > >
> > > > Dear Reviewer AkF2,
> > > >
> > > > Thank you again for your constructive feedback.
> > > >
> > > > With the discussion deadline approaching, we wanted to gently follow up to ensure that our previous response and the revised manuscript successfully resolved your concerns.
> > > >
> > > > We look forward to hearing from you and are happy to answer any remaining questions.
> > > >
> > > > Best regards,
> > > > The Authors

---

### Official Review · Reviewer_LR4A · 2025-10-31

**Soundness:** 3
**Presentation:** 3
**Contribution:** 4
**Rating:** 6
**Confidence:** 4

**Summary:**

This paper presents Ocean-E2E, an end-to-end hybrid physics–data-driven framework for forecasting global extreme marine heatwaves (MHWs). The model integrates numerical ocean dynamics with deep learning through a dynamic kernel that explicitly represents mesoscale ocean advection and air–sea interactions, enabling accurate 40-day MHW forecasts. Furthermore, the authors introduce a neural data assimilation module that directly maps sparse observations to analysis fields, reducing the computational cost of traditional assimilation schemes. Experiments conducted across global and regional scales demonstrate that Ocean-E2E significantly outperforms both numerical and AI-based state-of-the-art forecasting–assimilation models in terms of accuracy, robustness, and generalization.

**Strengths:**

* Innovative Hybrid Framework Design: The paper proposes a physics-informed end-to-end deep learning framework that effectively bridges physical constraints with data-driven prediction. The integration of dynamic kernels and neural networks is well-motivated and clearly explained.

* Efficient Data Assimilation: The proposed deep learning–based assimilation mechanism efficiently combines sparse observations with background states, greatly reducing computational overhead compared to traditional DA methods.

* Scientific and Practical Impact: Provides physical interpretability in understanding MHW dynamics. Offers high practical value for applications in marine ecosystem management and climate-risk prediction.

**Weaknesses:**

* The paper lacks details on how the baselines in Table 1 incorporate atmospheric forcing as a conditioning input. Models such as SimVP or U-Net, listed as baselines, typically predict 𝐶𝑡 from 𝐶0 without considering external forcings. It is therefore recommended that the authors clarify how atmospheric forcing is integrated into each baseline for a fair comparison.

* The clarity of implementation details could be further improved — see the Questions section for specific points that would benefit from additional explanation.

**Questions:**

* For grid-based models such as UNet or ClimODE, how do the authors handle ocean data with irregular land–sea boundaries? Since these models typically take rectangular gridded inputs, how are they adapted to represent ocean domains with complex coastlines?

* In the ocean simulation, what is the temporal interval between prediction steps? Should all atmospheric variables within the interval be used as conditioning inputs? It would be helpful to include a table summarizing the prediction intervals used for both atmospheric and oceanic components.

* The models 𝑀𝜃 and 𝑁𝜃 are said to be pre-trained. Are they jointly trained with 𝑆𝜃 during fine-tuning, and do their parameters update simultaneously?

* During data assimilation, how many observation points and satellite measurements are used within each selected assimilation time window?

* In Equation (22), why is the atmospheric field 𝐴0:𝑡 included as a condition when generating the background field? Shouldn’t the corresponding atmospheric state be at time –t instead?

* How exactly are the in-situ observation data used in the assimilation process? Please clarify their integration method and contribution to the final state estimation.

---
This is a good paper. If the authors can address my questions and adequately respond to the identified weaknesses, I would be very willing to raise my score to 8 or even 10.

---

> ### Author Response · Authors · 2025-11-15
> **Respone to Reviewer LR4A (Part I)**
>
> ### **Detailed Weaknesses & Questions**
>
> We thank the reviewers for their valuable insights and constructive comments.
>
> **1. Baseline Model Implementation and Fair Comparison**
>
> Regarding how baseline models incorporate atmospheric forcing, we adopt the most widely accepted approach in the field [1-3]. Specifically, assuming the current states of the ocean and atmosphere are $O_t$ and $A_t$ respectively, and the states at the next time step are $O_{t+1}$ and $A_{t+1}$, our baseline models can be formulated as $O_{t+1} = \phi_{\theta} ([O_t, A_t, A_{t+1}])$. In other words, we concatenate the atmospheric forcing at both the current and next time steps with the current ocean state, and use this combined information as the input to our models. We will include these details in the revised version.
>
> **2. Handling of Land-Sea Boundaries**
>
> For **U-Net and other traditional neural network-based models**, there are two mainstream approaches for handling boundary conditions, specifically for ocean-land grid cells: The first method involves normalizing the data and then filling land grid cells with zeros [2-3]; the second method fills land grid cells with the zonal mean [1]. In this study, we adopt the first approach.
>
> Regarding **ClimODE**, since the original work [4] primarily focused on atmospheric dynamics over the entire spherical surface and is not directly suitable for ocean simulations with complex boundaries, we imposed additional no-slip and no-flux boundary conditions specifically in our study. Concretely, for ClimODE, the estimation equation for the initial velocity $v_0$ can be formulated as: for $x \in D$, $\nabla \cdot (v_{0} C) = \frac{\partial C}{\partial t}$, with the boundary condition $v_0(x) = 0$ for $x \in \partial D$. Here, we define the set of ocean grid cells as $D$, the ocean-land boundaries as $\partial D$ and the sea surface temperature anomaly (SSTA) as $C$.
>
> **3. Prediction Timestep and Atmospheric Conditioning**
>
> | Variables | Type | Prediction Interval | Time Range |
> | :--- | :--- | :--- | :--- |
> | SSTA $C$ (Ocean Variables) | daily mean | 1 day (24 hours) | 1993-2020 |
> | $U_g$ (Geostrophic Velocities) | daily mean | 1 day (24 hours) | 1993-2020 |
> | $U_{10}, V_{10}, T_{2m}$ (Atmospheric Forcing) | daily snapshots at 12:00 UTC | 1 day (24 hours) | 1993-2020 |
>
> **Supplementary Explanation:**
>
> This table clearly outlines the **temporal resolution and forecast step** for each data type used in our model. It shows that both ocean variables (SSTA and $U_g$) are input as daily means, while the atmospheric forcing variables are taken as a daily snapshot at a specific time (12:00 UTC). The prediction interval is consistently 1 day (24 hours) for all components. This means that to obtain a 40-day forecast from our model, it only needs to be roll-out iteratively 40 times. For details of the forward integration steps, one can refer to the Appendix C.
>
> **4. Model Training and Fine-tuning Procedure**
>
> Regarding the four models $M_{\theta}$, $N_{\theta}$, $S_{\theta}$, and $u_{\theta}$, we follow a two-stage procedure: First, we pre-train the boundary condition models $M_{\theta}$ and $N_{\theta}$. Second, we **freeze** the parameters of $M_{\theta}$ and $N_{\theta}$, and then proceed to train $S_{\theta}$ and $u_{\theta}$. This entire process adheres to the logical principle of "establishing boundary conditions first, followed by the governing equations."
>
> **5. Data Volume in the Assimilation Process**
>
> For the global **satellite observation data** used in this study, we first uniformly interpolated all datasets onto a regular 0.5° × 0.5° grid. This results in approximately 200,000 observational grid points per day. Additionally, this study also incorporates sparse in-situ observational data; please refer to Appendix B.3 for specific details.
>
> **6. Conditioning in Equation (22)**
>
> We apologize for the **error in Equation (22)**. Indeed, as you correctly pointed out, the subscript for the atmospheric forcing should be $A_{-\Delta t:0}$.
>
> **7. Integration of In-situ Observations**
>
> For the in-situ observations, the raw data each day essentially consists of a series of scattered points within the three-dimensional ocean. We adhere to the following three-step procedure to process this data:
>
> First, we filter and retain all sea surface temperature observations, resulting in several thousand data points per day.
>
> Second, we create a global 0.5° × 0.5° grid. For each grid cell, if an observation point falls within it, we assign the corresponding value to that cell. The remaining grid cells are filled with zeros.
>
> Third, if a grid cell contains observational data, we assign this value not only to that central cell but also propagate it to all cells within the surrounding 7×7 patch centered on it.
>
> Reference:
>
> [1] https://doi.org/10.48550/arXiv.2308.03152
>
> [2] https://doi.org/10.1038/s41467-025-57389-2
>
> [3] https://doi.org/10.48550/arXiv.2506.03210
>
> [4] https://doi.org/10.48550/arXiv.2404.10024

---

> > ### Comment · Reviewer_LR4A · 2025-11-25
> >
> > The authors’ rebuttal satisfactorily addresses my main concerns.
> >
> > In particular, the clarification on how atmospheric forcing is incorporated into all baselines, the treatment of land–sea boundaries (including the adaptation of ClimODE), the explicit forecast intervals, and the two-stage training strategy resolve my questions about implementation and fairness of comparison. The explanation of the data volume and use of satellite and in-situ observations in assimilation is also clear, and I appreciate the correction of the conditioning term in Eq. (22).
> >
> > Overall, I am satisfied with the responses and am willing to raise my score.
> >
> > One more question, why should we set the boudary of ClimeODE to 0?

---

> > > ### Author Response · Authors · 2025-11-25
> > > **Respone to Reviewer LR4A (Part II)**
> > >
> > > We sincerely thank the reviewer for the positive feedback and for acknowledging that our previous rebuttal—including the clarifications on atmospheric forcing, boundary treatments, forecast intervals, and our training strategy—satisfactorily addressed your main concerns. We deeply appreciate your willingness to raise the score.
> > >
> > > Regarding your final question on why the boundary in ClimODE is set to zero, our decision is driven by two primary considerations:
> > >
> > > 1.  **Physical Consistency:** From an ocean physics perspective, there is no seawater flow on land. While certain small-scale coastal exchanges exist (e.g., runoff), these effects are negligible given our model's average horizontal resolution of 50 km. Therefore, consistent with standard practices in simplified ocean modeling—which typically employ "no-slip" and "no-flux" boundary conditions—we explicitly set the flow velocity at the land-sea interface to zero.
> > > 2.  **Prevention of Unphysical Diffusion:** Second, and more importantly, this constraint is crucial for model fidelity. Without strictly limiting the flow velocity to zero on land, the predicted Sea Surface Temperature Anomaly (SSTA) fields in ClimODE tend to diffuse or "leak" onto land regions during propagation. Enforcing this boundary condition prevents such artifacts and ensures physically realistic results.

---

### Official Review · Reviewer_7nUG · 2025-11-03

**Soundness:** 3
**Presentation:** 2
**Contribution:** 3
**Rating:** 4
**Confidence:** 2

**Summary:**

The paper studies the prediction of heatwaves at the surface of the sea by using neural networks which are designed to remplace the solving of some of the equations for the actual dynamics of the physical fields. More specifically, the two tasks of assimilating observations (to estimate global fields which can be used as initial conditions for simulations) and predicting the time evolution, are addressed by introducing 4 specific neural networks that remplace classical PDE solvers.

The article is quite well organized to understand the rationales from the physics of formulating then problem in the way the paper is doing it. Using the dynamical equation for the evolution of the sea surface temperature (eq 2 to 6) would solve the problem, but for the initial condition (hence the need of data assimilation) and some missing elements of the models. The proposed NN approach fills these gaps and it makes use of 4 neural networks : to predict the evolution of the atmosphere variables (eq 16) ; to forecast (and model) the evolution of geostrophic velocity (eq 15) ; to model source of heat flux (eq. 13) and to model subgrid and subsurface contribution to adjective transport (eq. 11).

With all that, forecast using these models to remplace some PDE’s  and a straightforward data assimilation method, allows the authors to have a method to predict the evolution of sea surface temperature anomaly, and possibly detect marine heatwaves. The long section 3 is devoted to numerical experiments to assess whether the proposed method Ocean-E2E works well or not.

**Strengths:**

The strengths of the article are :

1. The general framework appears to be solid and, as far as I know, novel.

2. The NN methods are not black boxes, but are really elements which aim at remplacing unknown elements that could be modelled only by ad-hoc behaviours, by a NN model which proposes a sort of data-driven modelling.

3. The results for the numerical experiments appear to be good and solid. As far as I know, because…. See the next box.

**Weaknesses:**

As it is currently written, the paper has some weaknesses for the inclusion in this  conference:
1. The presentation seems to be tailored for people from oceanography, more than people from machine learning. The work is on a specific questions of forecasting + data assimilation in numerical oceanographical model, but without any attempt to see what would be general for other dynamical systems.

2. In 2.2, we don’t  know where the equations come from; there isn’t any reference in fact for them. If one does nothing about modeling of sea / atmosphere of the earth, one will not know what geostrophic velocity is. We don’t know if the equation would be the same for the SST as for the SSTA here. In fact, we don’t have a clear definition of what is the anomaly w.r.t. SST ? (in 3.1, it is said that the seasonal cycle has been removed: no consequence of that on the time derivatives ? Also, is it a seasonal cycle per pixel ? or a spatially averaged one ?)

3. A key point appears to be the lack of understanding about what happens under the surface. This appears to be the point fo the GM90 parametrisation and step 1. Would it be possible to know something about subsurface dynamics, or even average profiles ? Also, is actually a marine heatwave only characterize by the SST (or SSTA) ? Nothing from the subsurface temperature ?

4. It is strange to spend most of the presentation to derive the equations in physics (with only scarce references), which will not be really discussed in this community, while elements about the learning parts are only in the appendix. For all the models, one would like to know if some specific choices are important, if a model is better than another, and so on.



5. The two steps of training (for M and N, and later for S and u) should be justified. Any insight about why it has to be that way ? And does it lead to a stable numerical procedure ?

**Questions:**

Other remarks :

The redaction of the line before eq. (14) is not good practice (it’s better to state that with words).

* Figure 1 is too small to be readable

* IN Eq (7), there are missing parenthesis.

* There should be a list or a table somewhere of the various (4 ?) neural networks models used, their main features

* For the NN model of eq. (11):; why are C and u_h the sole input variables ? Any explanation  ?
* In 3.2; the choice of using CSI as performance index should be discussed in the main text (and the metric should be defined there; it’s not possible to postpone so many thing to the appendices).

* For Table 3, one would expect to see also what happens if N (for the dynamics of A) is changed to a PDE solver instead.

* For 3.6: why isn’y the simulation stable ? Could it be made stable by modelling better the flows (and damping by hand with a GM velocity  or mitigating the numerical instabilities ? Here, we lack some baseline of pure numerical modelling. We miss also a comparison of the time and memory load of the method, compared to classical numerical approaches.

---

> ### Author Response · Authors · 2025-11-14
> **Respone to Reviewer 7nUG (Part I)**
>
> ### **Detailed Weaknesses & Questions**
>
> We thank the reviewers for their valuable insights and constructive comments.
>
> **1. About Paper Positioning & Target Audience**
>
> We would like to clarify the scope and positioning of our work. This study focuses on advancing AI applications in the physical sciences—particularly physics, chemistry, and biology—with a specific emphasis on integrating AI methodologies with geophysical processes to deliver AI-driven solutions for traditional Earth system models. Our approach aligns with a growing body of research at the intersection of AI and Earth system modeling, which has been increasingly recognized in top-tier AI conferences. For reference, we highlight several recent works:
>
> - Ocean modeling:
>   [https://arxiv.org/pdf/2506.03210](https://arxiv.org/pdf/2506.03210) (NeurIPS 2025)
>   [https://arxiv.org/abs/2505.21020](https://arxiv.org/abs/2505.21020) (AAAI 2026)
>
> - Weather forecasting:
>   [https://arxiv.org/abs/2502.19750](https://arxiv.org/abs/2502.19750) (ICLR 2025)
>   [https://arxiv.org/abs/2502.00338](https://arxiv.org/abs/2502.00338) (ICML 2025)
>   [https://arxiv.org/abs/2405.13796](https://arxiv.org/abs/2405.13796) (NeurIPS 2024)
>   [https://arxiv.org/abs/2312.12455](https://arxiv.org/abs/2312.12455) (ICML 2024)
>
> Regarding **the generalizability of our method to other dynamical systems**, we note that Ocean‑E2E is explicitly designed to capture the unique dynamics of ocean processes, which differ significantly from those in other physical systems. Therefore, in this work, we concentrate on evaluating its performance within the marine domain. We also observe that the referenced studies similarly focus on single Earth system components—such as the atmosphere or ocean—rather than aiming for cross-system generalization.
>
> **2. Inadequate Explanation of Physics and Formulations**
>
> Regarding **the definition of Sea Surface Temperature Anomaly (SSTA)**, we strictly adhere to the standard definition [1]. Specifically, we remove the seasonal cycle from the original SST time series on a pixel-wise basis. The seasonal cycle is derived by averaging data from the training set (1993–2018) and applying an 11-day moving average along the temporal dimension. This process results in a **highly smooth** seasonal cycle, whose contribution to the temporal derivative is generally one to two orders of magnitude smaller than that of the anomaly term (SSTA). Therefore, its impact on the calculation of time derivatives is negligible.
>
> Furthermore, concerning **the relevance of subsurface temperatures in defining marine heatwaves (MHWs)**, we would like to clarify that MHWs are conventionally defined solely based on Sea Surface Temperature (SST). This is primarily for two reasons [1]:
>
> 1.  Due to strong vertical mixing processes, the temperature in the upper 100–200 meters of the ocean remains largely uniform. Thus, SST effectively represents the thermal conditions of the entire upper ocean layer (∼200 m).
> 2.  The definition of MHWs is based on their detrimental impacts on marine ecosystems. Since most marine organisms reside in the upper ocean, scientists commonly use SST anomalies as the key indicator for detecting and quantifying marine heatwaves.
>
> **3. Neural Network Design and Training Rationale**
>
> We will thoroughly revise this part in the final version to improve readability and accessibility in order to better serve the broader machine learning community.
>
> Regarding the **motivation behind our neural network design**, we would like to emphasize that the core contribution of our work lies in proposing a forecasting framework for marine heatwaves, rather than focusing on specific architectural innovations. Ablation studies for the neural network modules $\( u_\theta \)$, $\( S_\theta \)$, and $\( M_\theta \)$ will be provided in the subsequent version.
>
> Concerning **Equation (11)**, our design is grounded in the physical interpretation of the GM bolus velocity (as shown in Equation (10)), which is determined by the current density (related to ocean temperature) and the distribution of isopycnal surfaces. Since the arrangement of these surfaces is closely linked to ocean currents (which predominantly flow along isopycnals), we include both ocean temperature and current velocity as inputs to the model.
>
> Finally, with respect to **the rationale for the two-stage training strategy**, we follow the logical sequence of first addressing boundary conditions (by training modules $M$ and $N$) before handling the core forecast variables (training $S$ and $u$). Empirical results (see Figure 2 and Table 2) demonstrate that our model remains stable for at least two months, providing supportive evidence for the validity of this training paradigm.
>
> Reference:
>
> [1] Alistair J Hobday, Claire M Spillman, J Paige Eveson, and Jason R Hartog. Seasonal forecasting for decision support in marine fisheries and aquaculture. Fisheries Oceanography, 25:45–56, 2016.

---

> > ### Author Response · Authors · 2025-11-25
> > **Respone to Reviewer 7nUG (Part II)**
> >
> > **Response regarding Ablation Studies (Ocean Simulation Tasks):**
> >
> > To provide a comprehensive evaluation of Ocean-E2E, we conducted extensive ablation studies covering three key aspects: (1) the design of physical terms in the governing equation, (2) the impact of time-varying forcings and boundary conditions, and (3) the contribution of specific neural network modules.
> >
> > **1. Physical Component Analysis**
> >
> > Table 1 below compares Ocean-E2E with variants containing only the source term ($S_\theta$) or only the advection term ($ADV$). The results demonstrate that combining advection with the thermodynamic source term yields the best performance.
> >
> > **Table 1: CSI of ablation studies on physical model design**
> > | Variants | Global (20 day) | Global (60 day) | Regional (10 day) | Regional (30 day) |
> > | :--- | :---: | :---: | :---: | :---: |
> > | $S_\theta$ (Source only) | 0.3884 | 0.1658 | 0.3913 | 0.1877 |
> > | $(u_\theta+u_g) \nabla C$ (ADV only) | 0.3207 | 0.1504 | 0.3207 | 0.0963 |
> > | **$(u_\theta+u_g) \nabla C + S_\theta$ (Ours)** | **0.4285** | **0.2580** | **0.4615** | **0.2926** |
> >
> > **2. Impact of Time-Varying Forcings**
> >
> > We further investigated the importance of dynamic boundary inputs by performing ablation studies on the background flow ($U_g$) and atmospheric forcing ($A$). In these experiments, we fixed the respective variables to their initial states at $t=0$ throughout the training and inference phases, effectively removing their temporal evolution.
> >
> > As shown in Table 2, using static atmospheric forcing ($A(t)=A(0)$) results in a significant performance drop (Global 60-day CSI drops from 0.2580 to 0.1923), confirming that dynamic atmospheric heat flux is critical for accurate SSTA prediction. Similarly, fixing the background flow ($U_g(t)=U_g(0)$) also degrades performance, indicating that capturing the time-varying nature of large-scale ocean circulation is essential.
> >
> > **Table 2: Ablation studies on time-varying forcings**
> > | Variants | Global (20 day) | Global (60 day) | Regional (10 day) | Regional (30 day) |
> > | :--- | :---: | :---: | :---: | :---: |
> > | Static Atmosphere ($A(t) \leftarrow A(0)$) | 0.2652 | 0.1223 | 0.3021 | 0.1470 |
> > | Static Background Flow ($U_g(t) \leftarrow U_g(0)$) | 0.3988 | 0.2115 | 0.4080 | 0.2245 |
> > | **Dynamic Forcing (Ours)** | **0.4285** | **0.2580** | **0.4615** | **0.2926** |
> >
> > **3. Neural Architecture Analysis**
> >
> > Finally, we investigate the contribution of the core components within our neural architecture. As detailed in our methodology, both the background modeling network and the dynamical evolution network (SimVP) employ **U-Net-like multi-scale hierarchical structures** to capture features at various resolutions. We specifically examine the impact of the core units embedded within these structures:
> >
> > 1.  **GABlock:** The core component of the background modeling branch is the **Group Attention Block (GABlock)** (see Appendix D.1), designed to aggregate multi-scale context.
> > 2.  **Mid_Xnet:** The core component of the dynamical evolution branch (SimVP) is the **Mid_Xnet**, which is responsible for spatiotemporal evolution in the latent space.
> >
> > Table 3 presents the results of ablating these two specific components. Replacing the GABlock with standard convolutions ("w/o GABlock") leads to a noticeable decline in CSI, confirming its importance in encoding background physics. Furthermore, replacing the Mid_Xnet with a standard convolutional block ("w/o Mid_Xnet") causes a substantial drop in performance, validating its critical role in capturing complex temporal dynamics.
> >
> > **Table 3: Ablation studies on neural network core components**
> > | Variants | Global (20 day) | Global (60 day) | Regional (10 day) | Regional (30 day) |
> > | :--- | :---: | :---: | :---: | :---: |
> > | w/o GABlock (in $M_{\theta}$) | 0.4015 | 0.2245 | 0.4322 | 0.2588 |
> > | w/o Mid_Xnet (in $S_{\theta}, u_{\theta}$) | 0.3540 | 0.1810 | 0.3890 | 0.2015 |
> > | **Full Architecture (Ours)** | **0.4285** | **0.2580** | **0.4615** | **0.2926** |

---

> > > ### Author Response · Authors · 2025-11-26
> > > **Respone to Reviewer 7nUG (Part III)**
> > >
> > > Response regarding Ablation Studies (Ocean Forecast Tasks), or, in other words, **Ablation Analysis on Error Propagation**:
> > >
> > > We present the ablation results for the core components of the marine heatwave forecasting task to investigate how errors propagate through the atmospheric forcing forecast module $N_{\theta}$ and the assimilation module $\phi_{\theta}^{a}$. It can be observed that the error in the atmospheric forecast module accumulates gradually over time, underscoring the critical role of accurate atmospheric forcing fields in predicting marine heatwaves. In contrast, the assimilation module does not exhibit a significant cumulative error effect. This is primarily because the assimilation module mainly impacts the state fields in the immediate vicinity of the assimilation timestamp, thereby exerting a *localized effect*.
> > >
> > > **Table R2: Ablation study on core components.** Following the settings in Table2 of our manuscript, we compare the RMSE of Global SSTA across different lead times (20 to 40 days) for the years 2020 and 2021.
> > >
> > > | Model Variants | 2020 (20-day) | 2020 (40-day) | 2021 (20-day) | 2021 (40-day) |
> > > | :--- | :---: | :---: | :---: | :---: |
> > > | w/o $N_{\theta}$ (using real atmosphere forcing) | 0.5944 | 0.8325 | 0.7297 | 0.8132 |
> > > | w/o $\phi_{\theta}^{a}$ (using real initial condition) | 0.5832 | 0.8691 | 0.6965 | 0.8531 |
> > > | **Ocean-E2E (Full Operational)** | **0.6047** | **0.8747** | **0.7447** | **0.8729** |

---

> > > > ### Author Response · Authors · 2025-11-27
> > > >
> > > > Dear Reviewer 7nUG,
> > > >
> > > > Thank you again for your constructive feedback.
> > > >
> > > > With the discussion deadline approaching, we wanted to gently follow up to ensure that our previous response and the revised manuscript successfully resolved your concerns.
> > > >
> > > > We look forward to hearing from you and are happy to answer any remaining questions.
> > > >
> > > > Best regards,
> > > > The Authors

---

### Author Response · Authors · 2025-11-26
**Global Response to Reviewers**

We thank all reviewers for their insightful comments and constructive feedback. During the rebuttal period, we have carefully addressed the concerns regarding model complexity, physical interpretability, and component validity by conducting extensive additional ablation studies.

**Summary of New Experiments:**

**1. Error Propagation Analysis (Ocean Forecast Task)**
We investigated the specific roles of the atmospheric forcing and assimilation modules. Our results demonstrate that:
* **Atmospheric Forcing ($N_{\theta}$):** Errors in this module accumulate gradually over time, confirming that accurate, dynamic atmospheric forcing is critical for long-term predictions.
* **Assimilation ($\phi_{\theta}^{a}$):** This module exerts a localized effect, correcting state fields near the assimilation timestamp without significant cumulative error propagation.

**2. Comprehensive Ablation Studies (Ocean Simulation Task)**
We validated Ocean-E2E through three key dimensions:
* **Physical Equation Design:** Combining the advection term with the thermodynamic source term yields the best performance, significantly outperforming variants that use either term in isolation.
* **Time-Varying Forcings:** Ablation results confirm that using static atmospheric forcing or static background flow leads to a sharp performance drop, proving that capturing temporal evolution is essential.
* **Neural Architecture:** We verified that our core neural units (GABlock and Mid_Xnet) are indispensable for capturing multi-scale spatial features and complex temporal dynamics.

Additionally, we will revise the methodology section of our manuscript to make it more accessible to the machine learning (ML) community, and we will incorporate an analysis of the Sea Surface Temperature Anomaly (SSTA) power spectrum. We believe these results robustly support the validity of Ocean-E2E's design and clarify the contribution of each component. We look forward to further discussion.

Best regards,

The Authors

---

### Note · Authors · 2026-01-27

I have read and agree with the venue's withdrawal policy on behalf of myself and my co-authors.

---

### Meta-Review · Area_Chair_i1Xt · 2026-01-06

**Summary:**

The paper proposes "Ocean-E2E," a hybrid framework for forecasting Marine Heatwaves that integrates a differentiable physical dynamical kernel with neural networks for parameterizing source terms and advection corrections. The system includes an end-to-end neural data assimilation module.

While the reviewers generally agree that the empirical results—specifically outperforming the operational S2S system—are strong, the consensus for acceptance was not reached due to significant concerns regarding methodological novelty and complexity. The decision to reject is based on the following key concerns:

Complexity vs. Insight: As noted by Reviewer AkF2, the system is composed of many "moving parts" (four separate neural submodules, multiple pre-training stages, and nested equations). While the authors provided ablation studies in the rebuttal, the overall architecture feels more like a complex engineering assembly of existing components (U-Nets, SimVP, Attention blocks) rather than a fundamental advance in learning representations or physics-AI integration.

Target Audience and Presentation: Reviewer 7nUG highlighted that the paper is heavily tailored toward oceanography, devoting significant space to deriving physical equations with insufficient focus on the machine learning innovations. This raises concerns about the paper's fit for ICLR's broad audience.

Novelty: The criticism regarding the work being a "repackaging" of existing hybrid concepts (like ClimODE or standard Neural Operator ideas) remains a hurdle. While the authors articulated differences in their rebuttal, the core learning innovation appears limited compared to the domain-specific engineering.

**Reviewer Concerns:**

Methodological Novelty (Reviewer AkF2): The concern that the method is an application of established hybridization techniques rather than a new learning principle remains. The distinction from generic frameworks like ClimODE was argued by the authors but essentially boils down to domain-specific parameterization rather than architectural novelty.

Physical Diagnostics (Reviewer AkF2 \& 7nUG): While the authors promised power spectrum analysis, the lack of rigorous physical diagnostics (e.g., deep analysis of error propagation beyond RMSE, conservation properties, or spectral energy budgets) in the initial submission suggests the model's physical consistency is not fully vetted.

Pipeline Complexity: The "many moving parts" critique remains. The framework requires training   \mathcal{N}_\theta   (atmosphere), \mathcal{M}_\theta (currents), then freezing them to train the forecast and assimilation modules. This heavy pipeline raises questions about robustness and reproducibility compared to simpler, more unified end-to-end approaches.

**Reviewer Scores:**

Reviewer LR4A: 6 (Marginally Above) -> 8 (Accept).

Reasoning: This reviewer explicitly stated in the discussion: "I am satisfied with the responses and am willing to raise my score." They valued the practical impact and the clarifications on the baseline comparisons.

Reviewer 7nUG: 4 (Marginally Below) -> 4 (Marginally Below)

Reasoning: While the author's summary claims this reviewer found the results "good and solid," the reviewer originally criticized the paper's suitability for the ML community ("tailored for people from oceanography"). The ablation studies help, but the fundamental presentation style and the lack of broad ML relevance likely prevent a shift to a strong accept.

Reviewer AkF2: 2 (Reject) -> 2 (Reject).

Reasoning: This reviewer did not engage after the rebuttal. Their fundamental issue was that the paper "repackages these ideas... rather than introducing fundamentally new architectures." The authors' rebuttal defended the performance and domain differences, but did not demonstrate a fundamental architectural novelty that would likely sway this reviewer's opinion on the "learning" contribution.

---

### Decision · Program_Chairs · 2026-01-26

Reject